# Impaired fatty acid import or catabolism in macrophages restricts intracellular growth of *Mycobacterium tuberculosis*

**Nelson V Simwela[1], Eleni Jaecklein[2], Christopher M Sassetti[2], David G Russell[1]***

[1]Department of Microbiology and Immunology, College of Veterinary Medicine, Cornell University, Ithaca, United States; [2]Department of Microbiology, UMass Chan Medical School, Worcester, United States

## eLife Assessment

This **important** study reveals that disrupting fatty acid metabolism in macrophages significantly restricts the growth of *Mycobacterium tuberculosis*, showing that impaired lipid processing triggers various antimicrobial responses. Overall, the approach is robust utilizing CRISPR-Cas9 knockout of multiple genes involved in lipid metabolism that yielded **convincing** data. This work highlights how host lipid metabolism affects the ability of tubercle bacilli to thrive intracellularly, pointing to potential new therapeutic targets.

***For correspondence:**
dgr8@cornell.edu

**Competing interest:** The authors declare that no competing interests exist.

**Abstract** *Mycobacterium tuberculosis* (*Mtb*) infection of macrophages reprograms cellular metabolism to promote lipid retention. While it is clearly known that intracellular *Mtb* utilize host-derived lipids to maintain infection, the role of macrophage lipid processing on the bacteria's ability to access the intracellular lipid pool remains undefined. We utilized a CRISPR-Cas9 genetic approach to assess the impact of sequential steps in fatty acid metabolism on the growth of intracellular *Mtb*. Our analyses demonstrate that macrophages that cannot either import, store, or catabolize fatty acids restrict *Mtb* growth by both common and divergent antimicrobial mechanisms, including increased glycolysis, increased oxidative stress, production of pro-inflammatory cytokines, enhanced autophagy, and nutrient limitation. We also show that impaired macrophage lipid droplet biogenesis is restrictive to *Mtb* replication, but increased induction of the same fails to rescue *Mtb* growth. Our work expands our understanding of how host fatty acid homeostasis impacts *Mtb* growth in the macrophage.

## Introduction

*Mycobacterium tuberculosis* (*Mtb*), the causative agent of tuberculosis (TB), has caused disease and death in humans for centuries (**WHO, 2023**). *Mtb* primarily infects macrophages in the lung (**Cohen et al., 2018**; **Wolf et al., 2007**), wherein the bacterium relies on host-derived fatty acids and cholesterol for the synthesis of its lipid-rich cell wall and to produce energy and virulence factors (**Peyron et al., 2008**; **Russell et al., 2009**; **Singh et al., 2012**; **Daniel et al., 2011**; **Muñoz-Elías and McKinney, 2005**; **Pandey and Sassetti, 2008**; **Brzostek et al., 2009**). Within the lung microenvironment, resident alveolar macrophages preferentially oxidize fatty acids and are more permissive to *Mtb* growth while recruited interstitial macrophages are more glycolytic and restrictive of *Mtb* replication (**Huang et al., 2018**; **Pisu et al., 2020**; **Pisu et al., 2021**). Globally, *Mtb* infection modifies macrophage metabolism in a manner that enhances its survival. *Mtb*-infected macrophages shift their mitochondrial substrate preference to exogenous fatty acids, which drives the formation of foamy macrophages that are laden

with cytosolic lipid droplets (*Peyron et al., 2008*; *Russell et al., 2009*; *Cumming et al., 2018*; *Singh et al., 2012*; *Podinovskaia et al., 2013*). Foamy macrophages are found in abundance in the central and inner layers of granulomas, a common histopathological feature of human TB (*Russell et al., 2009*; *Kim et al., 2010*). Interference with key regulators of lipid homeostasis, such as the miR-33 and the transcription factors peroxisome proliferator-activated receptor α (PPARα) and PPAR-γ, enhances macrophage control of *Mtb* (*Kim et al., 2017*; *Almeida et al., 2009*; *Ouimet et al., 2016*). More-over, compounds that modulate lipid metabolism such as the antidiabetic drug metformin and some cholesterol-lowering drugs are under investigation for host-directed therapeutics (HDTs) against *Mtb* (*Parihar et al., 2014*; *Singhal et al., 2014*). Although the dependence of intracellular *Mtb* on host fatty acids and cholesterol is well documented (*Wilburn et al., 2018*), the impact of specific aspects of macrophage lipid metabolism on the bacteria remains opaque. In *Mtb*-infected foamy macrophages, bacteria containing phagosomes are found in close apposition to intracellular lipid droplets (*Peyron et al., 2008*). It is believed that the bacterial induction of a foamy macrophage phenotype in host cells results in a steady supply of lipids that addresses the bacteria's nutritional requirements (*Peyron et al., 2008*; *Russell et al., 2009*; *Singh et al., 2012*; *Daniel et al., 2011*). In fact, intracellular *Mtb* has been shown to import fatty acids from host lipid droplet- derived triacylglycerols (*Daniel et al., 2011*). However, other studies indicate that macrophage lipid droplet formation in response to *Mtb* infection can lead to the induction of a protective, antimicrobial response (*Knight et al., 2018*). *Mtb* appears unable to acquire host lipids when lipid droplets are induced by stimulation with interferon gamma (IFN-γ) (*Knight et al., 2018*). Moreover, there is some evidence that lipid droplets can be sites for the production of host-protective pro-inflammatory eicosanoids (*Knight et al., 2018*; *Daniel et al., 2011*). Lipid droplets can also act as innate immune hubs against intracellular bacterial pathogens by clustering antibacterial proteins (*Bosch et al., 2020*). Inhibition of macrophage fatty acid oxidation by knocking out mitochondrial carnitine palmitoyl transferase 2 (CPT2) or using chemical inhibitors of CPT2 also restrict intracellular growth of *Mtb* (*Chandra et al., 2020*; *Huang et al., 2018*). These data demonstrate that modulation of the different stages in lipid metabolism inside *Mtb*-infected macro-phages can result in conflicting outcomes.

We carried out a candidate-based, CRISPR-mediated knockout of lipid import and metabolism genes in macrophages to determine their roles in intracellular growth of *Mtb*. By targeting genes involved in fatty acid import, sequestration, and metabolism in Hoxb8-derived conditionally immor-talized murine macrophages (*Kiritsy et al., 2021*), we show that impairing lipid homeostasis in macro-phages at different steps in the process negatively impacts the growth of intracellular *Mtb*, albeit to differing degrees. The impact on *Mtb* growth in the mutant macrophages was mediated through different mechanisms despite some common antimicrobial effectors. *Mtb*-infected macrophages defi-cient in the import of long-chain fatty acids increased the production of pro-inflammatory markers such as interleukin 1β (IL-1β). In contrast, ablation of lipid droplet biogenesis and fatty acid oxidation increased the production of reactive oxygen species (ROS) and limited the bacteria's access to nutri-ents. We also found that suppression of *Mtb* growth in macrophages that are unable to produce lipid droplets could not be rescued by exogenous addition of fatty acids, indicating that this is not purely nutritional restriction. Our data indicate that interference of lipid metabolism in macrophages leads to suppression of *Mtb* growth via multiple routes.

## Results

### Knockout of fatty acid import and metabolism genes restricts *Mtb* growth in macrophages

To apply a holistic approach to assessing the role(s) of fatty acid metabolism on the intracellular growth of *Mtb*, we used a CRISPR genetic approach to knockout genes involved in lipid import (CD36, SLC27A1), lipid droplet formation (PLIN2), and fatty acid oxidation (CPT1A, CPT2) in murine primary macrophages (*Figure 1A*). Deletion of CD36 or CPT2 in mouse macrophages has been shown to impair intracellular growth of *Mtb* (*Hawkes et al., 2010*; *Chandra et al., 2020*). But the role of special-ized long-chain fatty acid transporters (SLC27A1-6) on *Mtb* growth in macrophages is uncharacter-ized. SLC27A1 and SLC27A4 are the most abundant fatty acid transporter isoforms in macrophages (*Nishiyama et al., 2018*). PLIN2, or adipophilin, is known to be required for lipid droplet forma-tion (*Paul et al., 2008*; *Larigauderie et al., 2004*). Five isoforms of mammalian perilipins (PLIN) are

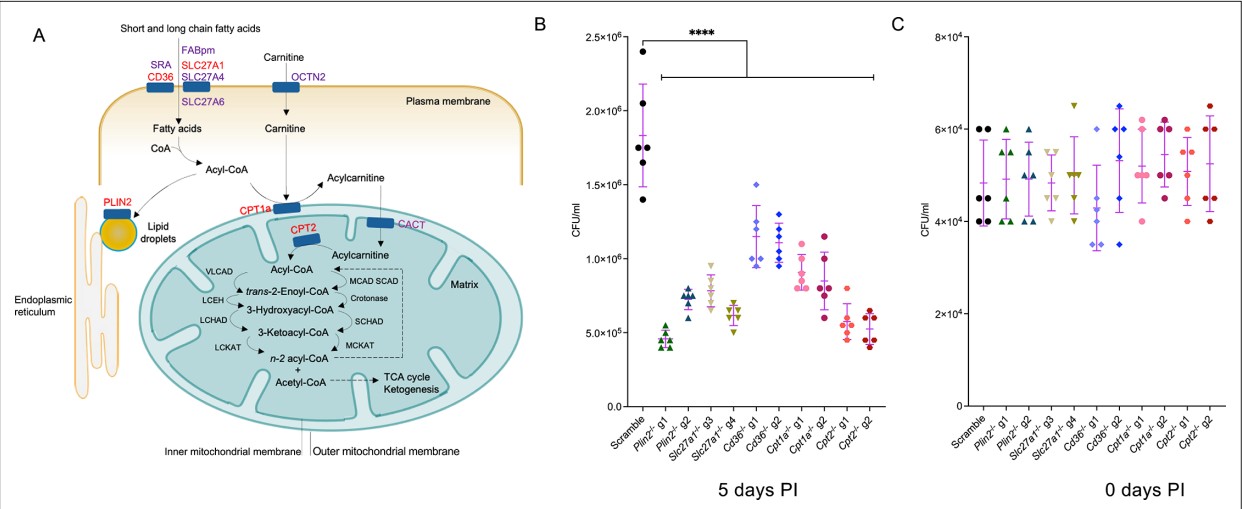

**Figure 1.** Knockout of fatty acid import and metabolism genes restricts *Mycobacterium tuberculosis* (*Mtb*) growth in macrophages. (**A**) Schematic of lipid import and metabolism genes in macrophages. Genes targeted for CRISPR-Cas9-mediated knockout are highlighted in red. (**B**) Scramble or indicated mutant macrophages were infected with the *Mtb* Erdman strain at a multiplicity of infection (MOI) of 0.4. Intracellular *Mtb* growth was measured by plating and counting colony-forming units (CFUs) in lysed macrophages 5 days post infection (PI). (**C**) CFUs from lysed macrophages were also plated on day 0, 3 hours PI to measure bacterial uptake differences. n = 6 biological replicates; ****p<0.0001, one-way ANOVA alongside Dunnett's multiple comparison test. Data are presented as mean values ±SD.

The online version of this article includes the following source data and figure supplement(s) for figure 1:

**Source data 1.** Numerical source data for *Figure 1B and C*.

**Figure supplement 1.** Flow cytometry and western blot analysis of CRISPR knockout macrophages.

**Figure supplement 1—source data 1.** Original western blots for *Figure 1—figure supplement 1A, B, D and E*, indicating the relevant bands.

**Figure supplement 1—source data 2.** Original files for western blot analysis displayed in *Figure 1—figure supplement 1A, B, D and E*.

**Figure supplement 2.** Lipid droplet formation and fatty acid oxidation in *Plin2⁻/⁻* and *Cpt2⁻/⁻* macrophages.

**Figure supplement 2—source data 1.** Numerical source data for *Figure 1—figure supplement 2B and C*.

---

involved in lipid droplet biogenesis among which PLIN2 is the dominant isoform expressed in macrophages (*Knight et al., 2018*). However, macrophages derived from *Plin2⁻/⁻* mice show no defects in the production of lipid droplets nor do they impair intracellular growth of *Mtb* (*Knight et al., 2018*). We targeted each of these genes with at least two sgRNAs in Hoxb8 Cas9⁺ conditionally immortalized myeloid progenitors (*Kiritsy et al., 2021*) to generate a panel of mutants that were deficient in the following candidates of interest; *Slc27a1⁻/⁻* p*lin2⁻/⁻*, *Cd36⁻/⁻*, *Cpt1a⁻*, and *Cpt2⁻*. Each individual sgRNA achieved >85% CRISPR-mediated deletion efficiency for all the five genes as analyzed by the Inference for CRISPR Edits (ICE) tool (*Conant et al., 2022*; *Supplementary file 1*). We verified the protein knockout phenotypes by flow cytometry and western blot analysis of differentiated macrophages derived from the CRISPR-deleted Hoxb8 myeloid precursors (*Figure 1—figure supplement 1*).

To confirm certain knockout phenotypes functionally, we checked lipid droplet biogenesis in *Plin2⁻/⁻* macrophages in comparison to macrophages transduced with a non-targeting scramble sgRNA by confocal microscopy of BODIPY-stained cells. Cells were cultured for 24 hours in the presence of exogenous oleate to enhance the formation of lipid droplets (*Listenberger and Brown, 2007*). We observed a complete absence of lipid droplet formation in *Plin2⁻/⁻* macrophages compared to controls (*Figure 1—figure supplement 2A*). This is contrary to previous observations in macrophages derived from PLIN2 knockout mice, which were reported to have no defect in lipid droplet formation (*Knight et al., 2018*). We also assessed the ability of *Cpt2⁻/⁻* macrophages to oxidize fatty acids using the Agilent Seahorse XF Palmitate Oxidation Stress Test. Scrambled sgRNA and *Cpt2⁻/⁻* macrophages were cultured in substrate-limiting conditions and supplied with either bovine serum albumin (BSA) or BSA-conjugated palmitate. As shown in *Figure 1—figure supplement 2B*, control macrophages were able to utilize and oxidize palmitate in substrate-limiting conditions indicated by a significant increase in oxygen consumption rates (OCRs) in contrast to cells supplied with BSA alone. Addition of the

CPT1A inhibitor, etomoxir, inhibited the cell's ability to use palmitate in these conditions (*Figure 1—figure supplement 2B*). CPT2 knockout in *Cpt2^-/-* macrophages impaired the cell's ability to oxidize palmitate to a degree comparable to etomoxir treatment as evidenced by baseline OCRs compared to scrambled sgRNA control (*Figure 1—figure supplement 2C*).

We then assessed the different knockout mutant macrophages in their ability to support the intracellular growth of *Mtb*. We infected macrophages with *Mtb* Erdman at a multiplicity of infection (MOI) of 0.4 and assessed intracellular bacterial growth rates by counting colony-forming units (CFUs) 5 days post infection. All the five mutant macrophages significantly impaired *Mtb* growth rates compared to scrambled sgRNA as assessed by CFUs counts on day 5 (*Figure 1B*). *Plin2^-/-*, *Slc27a1^-/-*, and *Cpt2^-/-* macrophages displayed the strongest growth restriction phenotypes while *Cd36^-/-* macrophages had a moderate, but significant, impact on *Mtb* growth. In parallel, we quantified intracellular bacteria on day 0, 3 hours post infection (*Figure 1C*), to ascertain that subsequent differences on day 5 were not due to disparities in initial bacterial uptake. The moderate growth restriction phenotypes of *Cd36^-/-* macrophages were consistent with previous findings, which reported a similar impact on *Mtb* and *M. marinum* growth in macrophages derived from *Cd36^-/-* mice (*Hawkes et al., 2010*). Impaired growth of *Mtb* in *Cpt1a^-/-* and *Cpt2^-/-* macrophages is also consistent with previous reports that genetic and chemical inhibition of fatty acid oxidation is detrimental to the growth of *Mtb* within macrophages (*Huang et al., 2018*; *Chandra et al., 2020*).

## *Mtb*-infected macrophages with impaired fatty acid import and metabolism display altered mitochondrial metabolism and elevated glycolysis

Impairment of fatty acids metabolism by SLC27A1 knockout in macrophages rewires their substrate bias from fatty acids to glucose (*Johnson et al., 2016*). We reasoned that deletion of genes required for downstream processing of lipids (*Figure 1A*) could also reprogram macrophages and increase glycolysis, which could, in part, explain bacterial growth restriction. We analyzed the metabolic states of three knockout macrophages (*Slc27a1^-/-*, *Plin2^-/-*, and *Cpt2^-/-*) in uninfected and *Mtb*-infected conditions by monitoring OCRs and extracellular acidification rates (ECARs) using the Agilent Mito and Glucose Stress Test kits. All three mutant uninfected macrophages displayed reduced mitochondrial respiration as evidenced by lower basal and spare respiratory capacity (SRC) compared to scrambled sgRNA controls (*Figure 2—figure supplement 1A and B*). *Mtb* infection proportionally reduced basal and SRC rates across all the mutant macrophages and scrambled controls (*Figure 2A and B*) compared to uninfected macrophages, which is consistent with previous findings (*Cumming et al., 2018*). *Plin2^-/-* macrophages displayed the most marked reduction in mitochondrial activity in both uninfected and infected conditions, while *Slc27a1^-/-* macrophages were the least affected (*Figure 2A*, *Figure 2—figure supplement 1A*). As reported previously (*Johnson et al., 2016*), uninfected *Slc27a1^-/-* macrophages were more glycolytically active with higher basal and spare glycolytic capacity (SGC) compared to scrambled controls (*Figure 2—figure supplement 1C and D*). Uninfected *Plin2^-/-* and *Cpt2^-/-* macrophages were also more glycolytically active, but to a greater degree than *Slc27a1^-/-* (*Figure 2—figure supplement 1C and D*). *Mtb* infection increased the glycolytic capacity of all the three mutant macrophages (*Figure 2C and D*). Overall, *Plin2^-/-* macrophages exhibited the highest glycolytic capacity (*Figure 2C*, *Figure 2—figure supplement 1D*). Our data indicates that impairment of fatty acid metabolism at different steps significantly impacts mitochondrial respiration and reprograms cells toward glycolysis. Increased glycolytic flux in macrophages has been linked to the control of intracellular *Mtb* growth (*Gleeson et al., 2016*; *Shi et al., 2015*). Metabolic realignment as a consequence of interference with lipid homeostasis, which results in enhanced glycolysis may contribute to *Mtb* growth restriction in these mutant macrophages.

## Knockout of lipid import and metabolism genes in macrophages activates AMPK and stabilizes HIF1α

*Mtb* infection is known to induce increased glycolysis or the 'Warburg effect' in macrophages, mouse lungs, and human TB granulomas (*Shi et al., 2015*; *Gleeson et al., 2016*; *Belton et al., 2016*). Several studies have demonstrated that the Warburg effect is mediated by the master transcription factor hypoxia-inducible factor 1 (HIF1) (*Courtnay et al., 2015*). During *Mtb* infection, HIF1 is activated by the production of ROS, tricarboxylic cycle (TCA) intermediates and hypoxia in the cellular

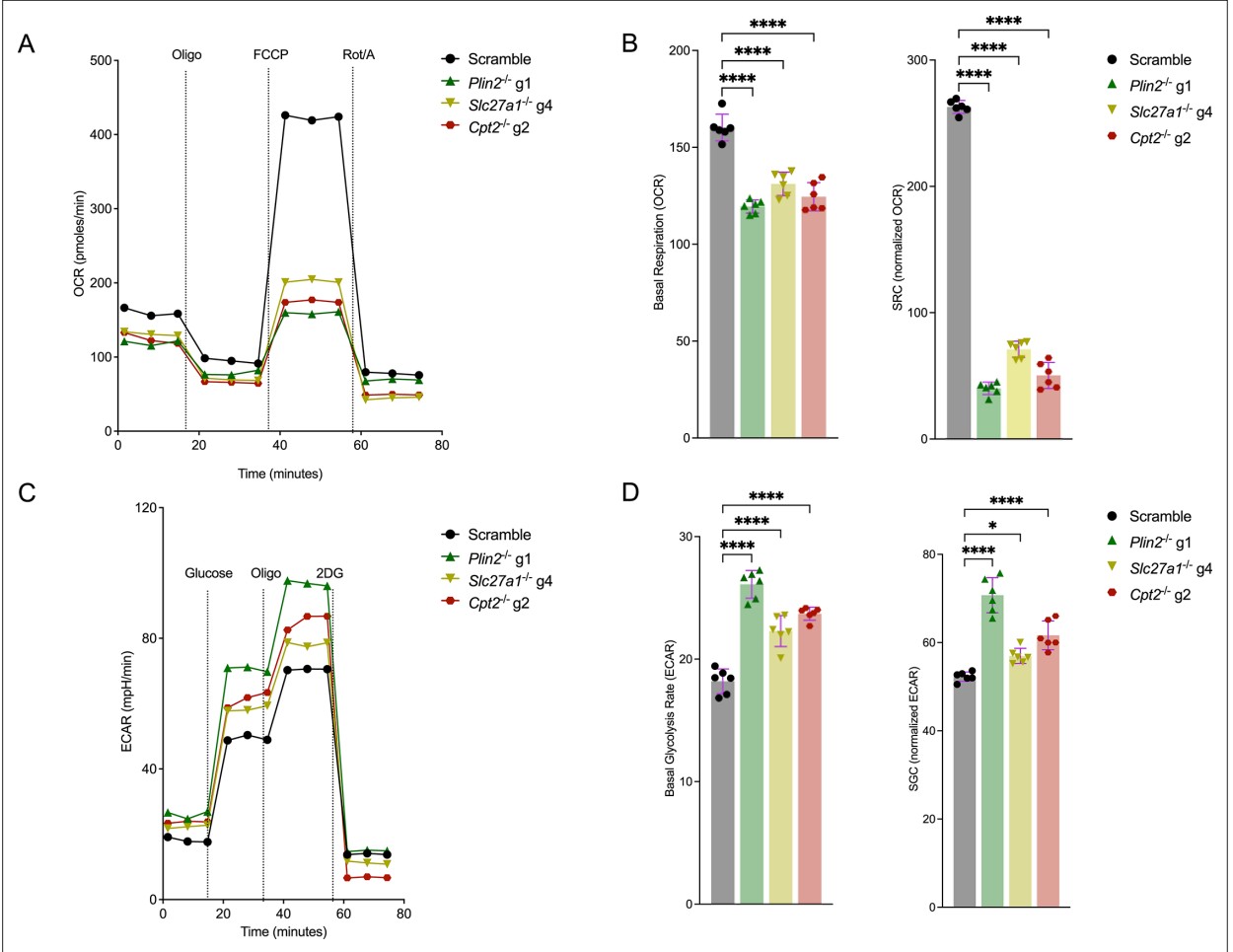

**Figure 2.** *Mycobacterium tuberculosis* (*Mtb*)-infected macrophages with impaired fatty acid import and metabolism display reduced mitochondrial activities and are more glycolytic. (**A**) Seahorse flux analyses of scramble or *Plin2*⁻/⁻, *Slc27a1*⁻/⁻, and *Cpt2*⁻/⁻ macrophages infected with *Mtb* Erdman strain at a multiplicity of infection (MOI) of 1 24 hours post infection. Oxygen consumption rates (OCRs) were measured using the Cell Mito Stress Test Kit (Agilent). Oligo, oligomycin; FCCP, fluoro-carbonyl cyanide phenylhydrazone; Rot/A, rotenone and antimycin A. (**B**) Comparison of basal respiration and spare respiratory capacity (SRC) from (**A**). SRC was calculated by subtracting the normalized maximal OCR from basal OCR. n = 3 biological replicates (two technical repeats per replicate); ****p<0.0001, one-way ANOVA alongside Dunnett's multiple comparison test. Data are presented as mean values ± SD. (**C**) Extracellular acidification rates (ECARs) of scramble or *Plin2*⁻/⁻, *Slc27a1*⁻/⁻, and *Cpt2*⁻/⁻ macrophages infected with *Mtb* as in (**A**). ECARs were measured using the Agilent Seahorse Glycolysis Stress Test kit. 2DG, 2-deoxy-ᴅ-glucose. (**D**) Comparison of basal glycolysis and spare glycolytic capacity (SGC) in the indicated mutant macrophages. SGC was calculated as SRC above. n = 3 biological replicates (two technical repeats per replicate); *p<0.05, ****p<0.0001, one-way ANOVA alongside Dunnett's multiple comparison test. Data are presented as mean values ± SD.

The online version of this article includes the following source data and figure supplement(s) for figure 2:

**Source data 1.** Numerical source data for *Figure 2A–D*.

**Figure supplement 1.** Flux analyses of mitochondrial activities and glycolysis in *Plin2*⁻/⁻, *Slc27a1*⁻/⁻, and *Cpt2*⁻/⁻ macrophages.

**Figure supplement 1—source data 1.** Numerical source data for *Figure 2—figure supplement 1A–D*.

microenvironments as a consequence of altered metabolic activities and increased immune cell functions (*Li et al., 2024*; *Shi et al., 2015*; *Gleeson et al., 2016*; *Belton et al., 2016*). We assessed HIF1 stability in the three mutant macrophage lineages (*Slc27a1*⁻/⁻, *Plin2*⁻/⁻, and *Cpt2*⁻/⁻) by monitoring total HIF1α protein levels by western blot, having confirmed that they were all more glycolytically active than the scrambled controls (*Figure 2*, *Figure 2—figure supplement 1*). Indeed, all the three mutant macrophages displayed significantly higher amounts of total HIF1α compared to scrambled controls after *Mtb* infection for 48 hours (*Figure 3A*, *Figure 3—figure supplement 1A and B*). *Slc27a1*⁻/⁻, *Plin2*⁻/⁻ macrophages had increased levels of total HIF1α even in uninfected states (*Figure 3A and B*, *Figure 3—figure supplement 1A and B*). We also checked the phosphorylation status of the

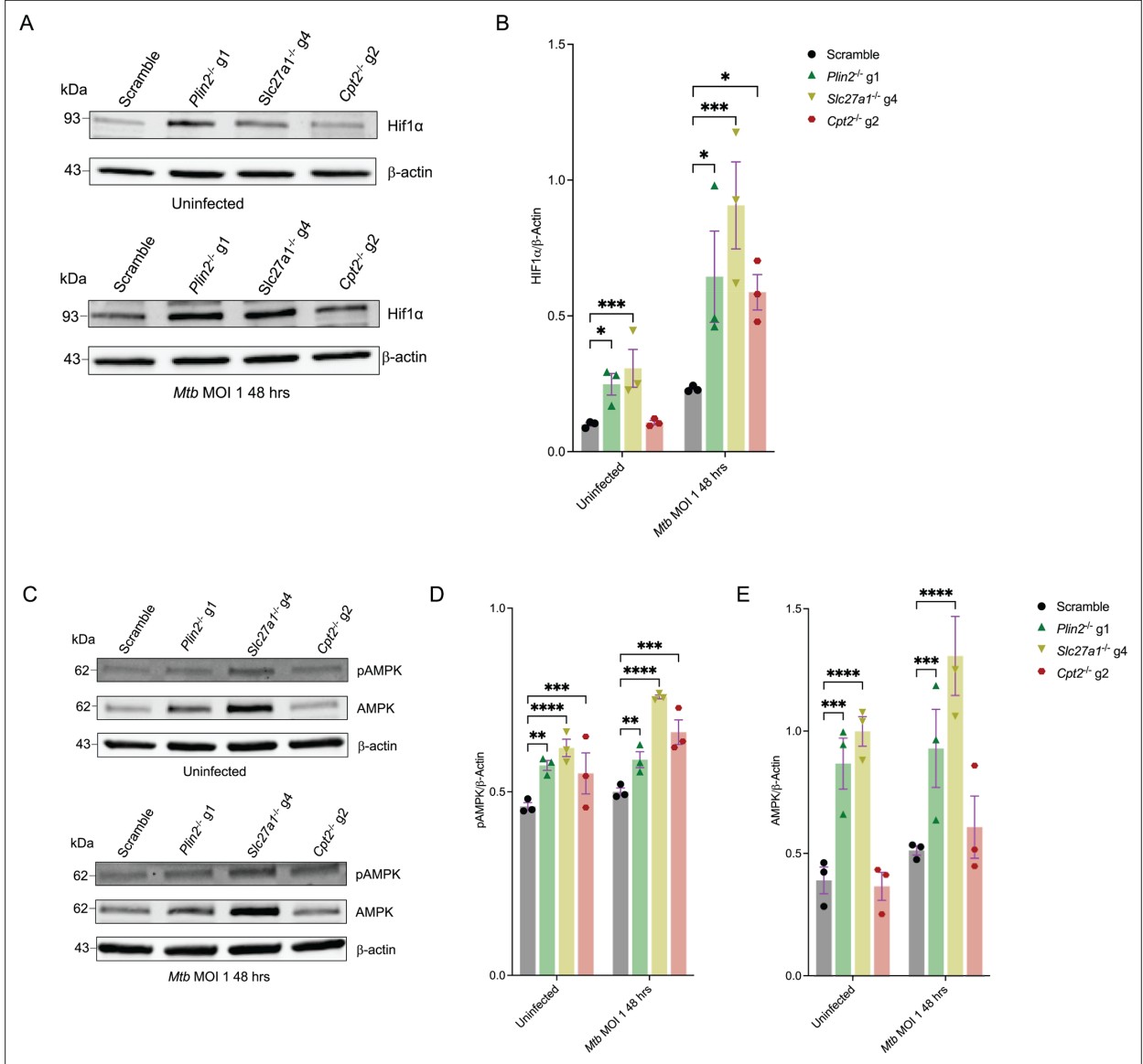

**Figure 3.** Knockout of fatty acid import and metabolism genes in macrophages activate AMPK and stabilizes HIF1α. (**A**) Western blot analysis of HIF1α in uninfected and *Mycobacterium tuberculosis* (*Mtb*)-infected scramble or *Plin2*−/−, *Slc27a1*−/−, and *Cpt2*−/− macrophages. In *Mtb*-infected conditions, cells were infected with the bacteria at a multiplicity of infection (MOI) of 1 for 48 hours before preparation of cell lysates. (**B**) Quantification of relative expression of HIF1α in (**A**) and in *Figure 3—figure supplement 1A and B* normalized to β-actin; n = 3 biological replicates. *p<0.05; ***p<0.001, two-way ANOVA alongside Dunnett's multiple comparison test. Data are presented as mean values ± SD. (**C**) Western blot analysis of total and phosphorylated AMPK in uninfected and *Mtb*-infected scramble or *Plin2*−/−, *Slc27a1*−/−, and *Cpt2*−/− macrophages. Cell lysates were prepared as in (**A**). (**D, E**) Quantification of relative expression of pAMPK and AMPK in (**C**) and in *Figure 3—figure supplement 1C and D* normalized to β-actin; n = 3 biological replicates. **p<0.01; ***p<0.001, ****p<0.0001, two-way ANOVA alongside Dunnett's multiple comparison test. Data are presented as mean values ± SD.

The online version of this article includes the following source data and figure supplement(s) for figure 3:

**Source data 1.** Original western blots for *Figure 3A and C*, indicating the relevant bands.

**Source data 2.** Original files for western blot analysis displayed in *Figure 3A and C*.

**Source data 3.** Numerical source data for *Figure 3B, D, and E*.

**Figure supplement 1.** Replicate western blot analyses of HIF1α (**A, B**), AMPK and pAMPK (**C, D**) in uninfected and *Mtb*-infected scramble or *Plin2*−/−, *Slc27a1*−/−, and *Cpt2*−/− macrophages as in *Figure 3*.

**Figure supplement 1—source data 1.** Original western blots for *Figure 3—figure supplement 1A–D*, indicating the relevant bands.

**Figure supplement 1—source data 2.** Original files for western blot analysis displayed in *Figure 3—figure supplement 1A–D*.

*Figure 3 continued on next page*

*Figure 3 continued*

**Figure supplement 2.** Increased autophagy in *Plin2⁻/⁻*, *Slc27a1⁻/⁻*, and *Cpt2⁻/⁻* macrophages.

**Figure supplement 2—source data 1.** Original western blots for *Figure 3—figure supplement 2A and B*, indicating the relevant bands.

**Figure supplement 2—source data 2.** Original files for western blot analysis displayed in *Figure 3—figure supplement 2A and B*.

**Figure supplement 2—source data 3.** Numerical source data for *Figure 3—figure supplement 2C, D, and E*.

adenosine monophosphate kinase (AMPK), a master regulator of cell energy homeostasis (*Garcia and Shaw, 2017*), in the mutant macrophages since Seahorse flux analyses indicated that they had impaired mitochondrial activities (*Figure 2*, *Figure 2—figure supplement 1*). Western blot analysis revealed that in both *Mtb*-infected and uninfected conditions, impaired fatty acid metabolism in the mutant macrophages correlated with increased activation of AMPK as indicated by higher levels of phosphorylated AMPK compared to scramble (*Figure 3C and D*, *Figure 3—figure supplement 1C and D*). Interestingly, total AMPK was also increased, at least in *Slc27a1⁻/⁻* and *Plin2⁻/⁻* macrophages, in both uninfected and *Mtb*-infected conditions (*Figure 3C and E*, *Figure 3—figure supplement 1C and D*). These data point to a metabolic reprogramming of cells through activation of HIF1α and AMPK to promote glycolysis. In energetically stressed cellular environments, activated AMPK promotes catabolic processes such as autophagy to maintain nutrient supply and energy homeostasis (*Garcia and Shaw, 2017*). Autophagy is also an innate immune defense mechanism against intracellular *Mtb* in macrophages (*Gutierrez et al., 2004*). We examined the levels of autophagic flux in the mutant macrophages by monitoring LC3I to LC3II conversion by western blot and by qPCR analysis of selected autophagy genes (AMBRA1, ATG7, MAP1LC3B and ULK1). We observed an increase in autophagic flux by western blot analysis of LC3II/LC3I ratios in *Mtb*-infected *Slc27a1⁻/⁻* and *Cpt2⁻/⁻* macrophages (*Figure 3—figure supplement 2A–C*). *Cpt2⁻/⁻* macrophages were more autophagic even in uninfected conditions (*Figure 3—figure supplement 2A–C*). Meanwhile, our qPCR analysis also revealed that the four autophagy genes were upregulated in both *Mtb*-infected and uninfected conditions in all the three mutants (*Figure 3—figure supplement 2D and E*). These data suggest that impaired fatty acid import and metabolism in macrophages could be restricting *Mtb* growth by promoting autophagy. These data agree with previous observations that inhibition of fatty acid oxidation enhances macrophage xenophagic activity, which leads to improved control of *Mtb* (*Chandra et al., 2020*).

## Exogenous oleate fails to rescue the *Mtb icl1*-deficient mutant in *Slc27a1⁻/⁻*, *Plin2⁻/⁻*, and *Cpt2⁻/⁻* macrophages

The mycobacterial isocitrate lyase (*icl1*) acts as an isocitrate lyase in the glyoxylate shunt and as a methyl-isocitrate lyase in the methyl-citrate cycle (MCC) (*Gould et al., 2006*; *McKinney et al., 2000*). *Mtb* uses the MCC to convert propionyl CoA originating from the breakdown of cholesterol rings and β-oxidation of odd chain fatty acids into succinate and pyruvate, which are eventually assimilated into the tricarboxylic cycle (TCA) (*Muñoz-Elías et al., 2006*; *Griffin et al., 2012*). The buildup of propionyl CoA is toxic to *Mtb* and the bacteria relies on the MCC together with the incorporation of propionyl CoA to methyl-branched lipids in the cell wall as an internal detoxification system (*Muñoz-Elías et al., 2006*; *Savvi et al., 2008*). *Mtb* propionyl CoA toxicity is, in part, due to a cellular imbalance between propionyl CoA and acetyl CoA as an accumulation of the former or paucity of the latter results in the propionyl CoA-mediated inhibition of pyruvate dehydrogenase (*Lee et al., 2013*). Consequently, *Mtb icl1*-deficient mutants (*Mtb Δicl1*) are unable to grow in media supplemented with cholesterol or propionate, or intracellularly in macrophages (*Lee et al., 2013*). However, this growth inhibition could be rescued both in culture and in macrophages by exogenous supply of acetate or even chain fatty acids, which can be oxidized to acetyl-CoA (*Lee et al., 2013*). We took advantage of this metabolic knowledge to assess whether exogenous addition of the even chain fatty acid oleate can rescue the intracellular growth of *Mtb Δicl1* mutants in our CRISPR knockout macrophages. Scrambled controls, *Slc27a1⁻/⁻*, *Plin2⁻/⁻*, and *Cpt2⁻/⁻* macrophages in normal macrophage media or media supplemented with oleate were infected with an *Mtb Δicl1* strain expressing mCherry at MOI 5. Bacterial growth measured by mCherry expression was recorded 5 days post infection. Consistent with previous observations (*Lee et al., 2013*), the *Mtb Δicl1* mutant failed to replicate in both mutant and scramble macrophages that were grown in normal macrophage media as evidenced by baseline mCherry fluorescence

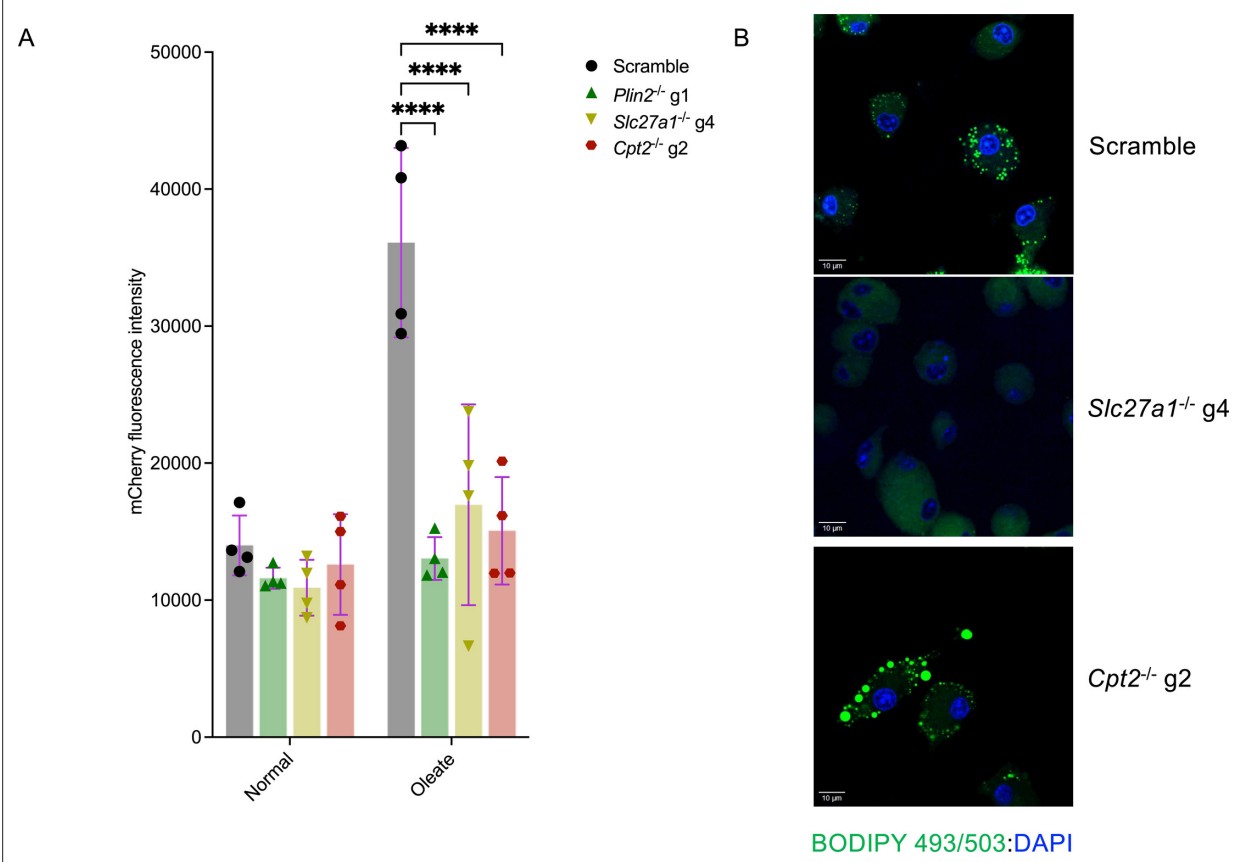

**Figure 4.** Supplementation with exogenous oleate fails to rescue the *Mycobacterium tuberculosis* (*Mtb*) Δicl1 mutant in *Plin2*[-/-], *Slc27a1*[-/-], and *Cpt2*[-/-] macrophages. (**A**) Scramble or indicated mutant macrophages were infected with the *Mtb* H37Rv Δicl1 mutant expressing mCherry at a multiplicity of infection (MOI) of 5. Oleate supplementation (400 µM) was commenced 24 hours before infection in the treatment group, removed during *Mtb* infection and readded 3 hours post infection for the entire duration of the experiment. Growth kinetics of *Mtb* were measured by monitoring mCherry fluorescence using a plate reader. n = 4 biological replicates; ****p<0.0001, two-way ANOVA alongside Dunnett's multiple comparison test. Data are presented as mean values ± SD. (**B**) Uninfected scramble or *Slc27a1*[-/-] and *Cpt2*[-/-] macrophages were supplemented with 400 µM oleate for 24 hours. Cells were then fixed for 20 minutes and stained for lipid droplet inclusions using the Bodipy 493/503 dye. DAPI was used as a counterstain to detect nuclei.

The online version of this article includes the following source data for figure 4:

**Source data 1.** Numerical source data for *Figure 4A*.

(*Figure 4A*). Oleate supplementation successfully rescued the *Mtb Δicl1* mutant in scrambled control macrophages. However, the growth restriction of the *Mtb Δicl1* strain could not be alleviated by exogenous addition of oleate to the mutant macrophages (*Slc27a1*[-/-], *Plin2*[-/-], and *Cpt2*[-/-]) (*Figure 4A*). These data suggest that impaired import (*Slc27a1*[-/-]), sequestration (*Plin2*[-/-]), or β-oxidation of fatty acids (*Cpt2*[-/-]) blocks *Mtb*'s ability to access and use cellular lipids.

Oleate supplementation in macrophages induces the formation of lipid droplets, and we were able to confirm the inability to produce lipid droplets in *Plin2*[-/-] macrophages using this approach (*Figure 1—figure supplement 2A*). As an indirect measure to track the fate of supplemented oleate in the mutant macrophages, we monitored lipid droplet biogenesis in *Slc27a1*[-/-] and *Cpt2*[-/-] macrophages to check if the inability to rescue the *Mtb Δicl1* impaired growth phenotypes in these mutant macrophages could be possibly related to disruptions in lipid droplet formation. Confocal analysis of BODIPY-stained cells upon oleate supplementation revealed that *Slc27a1*[-/-] macrophages also fail to generate lipid droplets (*Figure 4B*). In contrast, *Cpt2*[-/-] macrophages produced more and larger lipid droplets in comparison to scrambled controls (*Figure 4B*). These data suggest that the inhibition of *Mtb* growth in these mutant macrophages is not merely through limitation of access to fatty acid nutrients.

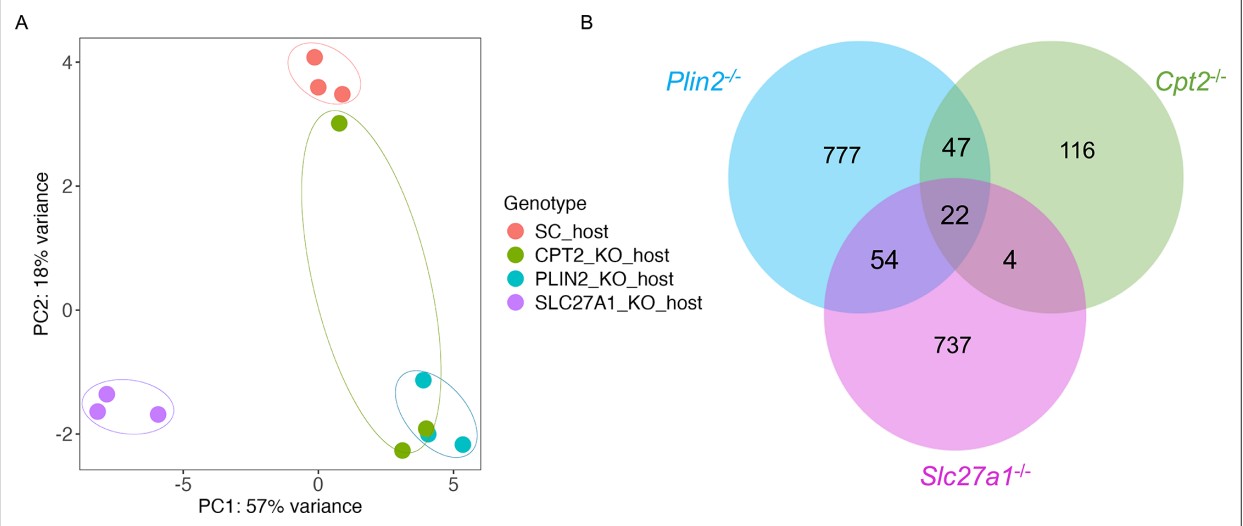

**Figure 5.** Dual RNA sequencing to identify host and bacterial determinants of *Mycobacterium tuberculosis* (*Mtb*) restriction in macrophages with fatty acid import and metabolism knockout genes. (**A**) Principal component analysis (PCA) of scramble or *Plin2*[-/-], *Slc27a1*[-/-], and *Cpt2*[-/-] macrophages transcriptomes infected with the *Mtb* smyc'::mCherry strain at a multiplicity of infection (MOI) of 0.5 4 days post infection. (**B**) Venn diagram of differentially expressed (DE) gene sets (*Supplementary file 2*) in *Plin2*[-/-], *Slc27a1*[-/-], and *Cpt2*[-/-] mutant macrophages compared to scramble showing overlapping genes. DE genes cutoff; abs (log$_2$ fold change) > 0.3, adjusted p-value<0.05.

## Dual RNA sequencing to identify host and bacterial determinants of *Mtb* restriction in mutant macrophage lineages

We performed RNA sequencing of both host and bacteria in *Mtb*-infected mutant macrophages as a preliminary step in the identification of pathways restricting bacterial growth. We infected scrambled controls, *Slc27a1*[-/-], *Plin2*[-/-], and *Cpt2*[-/-] macrophages with the *Mtb* smyc'::mCherry strain for 4 days and processed the samples for dual RNA sequencing as previously described (*Simwela et al., 2024*). Principal component analysis (PCA) of host transcriptomes revealed a clustering of all the three mutant macrophages away from scrambled controls (*Figure 5A*). Interestingly, there was a separation in transcriptional responses between the three mutant macrophages as *Cpt2*[-/-] and *Plin2*[-/-] macrophages clustered closer together and more distant from *Slc27a1*[-/-] (*Figure 5A*). Overall, using an adjusted p-value<0.05 and absolute log$_2$ fold change >1.2, we identified 900 genes that were differentially expressed (DE) in *Plin2*[-/-] macrophages (589 up, 311 down), 817 genes that were DE in *Slc27a1*[-/-] macrophages (501 up, 315 down), and 189 genes that were DE in *Cpt2*[-/-] (124 up, 65 down) (*Supplementary file 2*, *Figure 5B*). Consistent with the PCA, Venn diagram of the DE genes (*Figure 5B*) indicated divergent responses in the three mutant macrophage populations. We performed pathway enrichment analysis (*Wu et al., 2021*) of the DE genes to identify antimicrobial pathway candidates in the three mutant macrophages. We found that defects in fatty acid uptake in *Slc27a1*[-/-]-infected macrophages upregulated pro-inflammatory pathways involved in MAPK and ERK signaling and production of inflammatory cytokines (IFN-γ, IL-6, IL-1α, β) (*Figure 6A*, *Supplementary file 3*). The pro-inflammatory signatures of the *Slc27a1*[-/-] macrophages are consistent with previous observations that demonstrated that a deficiency in *Slc27a1*[-/-] exacerbated macrophage activation in vitro and in vivo (*Johnson et al., 2016*). SLC27A1 is a solute carrier family member transporter and *Mtb*-infected *Slc27a1*[-/-] macrophages showed reduced expression of other solute carrier transporters (SLC) such as SLC27A4, GLUT1 (SLC2A1), and eight SLC amino acid transporters (*Figure 6—figure supplement 1A, B and 2A*, *Supplementary file 2*). Interestingly, *Mtb*-infected *Slc27a1*[-/-] macrophage transcriptomes exhibited upregulation of macrophage scavenger receptors (MSR1) and the ATP binding cassette transporter ABCC1 (*Figure 6—figure supplement 1A*), both of which can independently transport fatty acids into cells (*Vogel et al., 2022*; *Raggers et al., 1999*).

Meanwhile, *Mtb* infection of *Plin2*[-/-] macrophages led to upregulation in pathways involved in ribosomal biology, MHC class 1 antigen presentation, canonical glycolysis, ATP metabolic processes, and type 1 interferon responses (*Figure 6B*, *Supplementary file 3*). In the downregulated *Plin2*[-/-] DE gene set, enriched pathways included those involved in the production of pro-inflammatory cytokines; IL-6

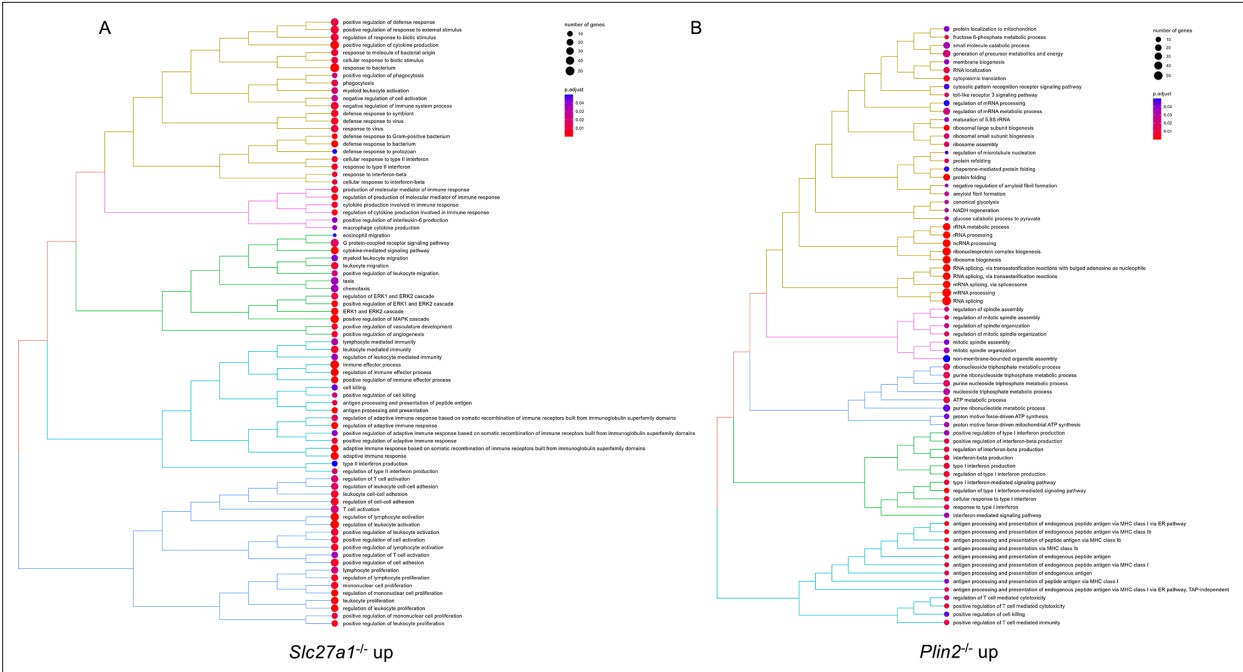

**Figure 6.** Pathway enrichment analysis of upregulated genes in *Mycobacterium tuberculosis* (*Mtb*)-infected *Plin2*-/- and *Slc27a1*-/- macrophages. Tree plots of top 80 enriched gene ontology terms (biological process) in *Mtb*-infected *Slc27a1*-/- (**A**) and *Plin2*-/- (**B**) upregulated genes.

The online version of this article includes the following source data and figure supplement(s) for figure 6:

**Figure supplement 1.** Compensatory transcriptional responses in *Mycobacterium tuberculosis* (*Mtb*)-infected *Slc27a1*-/- macrophages.

**Figure supplement 2.** Tree plot of top 80 enriched gene ontology terms (biological process) in *Slc27a1*-/- (A) and *Plin2*-/- (B) macrophages downregulated genes.

**Figure supplement 3.** Tree plot of top 80 enriched gene ontology terms (biological process) in *Cpt2*-/- macrophages upregulated genes.

**Figure supplement 4.** qPCR analysis of IFN-β (A) and IL-1β (B) in *Plin2*-/-, *Slc27a1*-/-, and *Cpt2*-/- macrophages.

**Figure supplement 4—source data 1.** Numerical source data for *Figure 6—figure supplement 4*.

and 8, IFN-γ and IL-1 (*Figure 6—figure supplement 2B*, *Supplementary file 3*). Oxidative phosphorylation and processes involved in the respiratory chain electron transport were also significantly enriched in *Plin2*-/- downregulated genes. This suggests that *Mtb*-infected *Plin2*-/- macrophages increase glycolytic flux and decrease mitochondrial activities, which is consistent with our metabolic flux analysis data (*Figure 2*, *Figure 2—figure supplement 1*). Unlike *Slc27a1*-/- macrophages, *Plin2*-/- macrophages are, however, broadly anti-inflammatory as most pro-inflammatory genes were downregulated upon *Mtb* infection.

Further downstream in the lipid processing steps, inhibition of fatty acid oxidation in *Cpt2*-/- macrophages upregulated pathways involved in MHC class 1 antigen presentation, response to IFN-γ and IL-1 and T-cell-mediated immunity (*Figure 6—figure supplement 3*, *Supplementary file 3*). There was a limited overlap in enriched pathways in the upregulated genes between *Mtb*-infected *Cpt2*-/- and *Slc27a1*-/- macrophages such as those involved in the cellular responses to IL-1 and IFN-γ. However, many pathways over-represented in *Cpt2*-/- macrophages were common to *Plin2*-/- macrophages (*Figure 6B*, *Figure 6—figure supplement 3*, *Supplementary file 3*). Similarly, both *Mtb*-infected *Plin2*-/- and *Cpt2*-/- macrophages were downregulated in the expression of genes involved in oxidative phosphorylation (*Supplementary file 3*). We confirmed expression levels of a selected gene by qPCR analysis of IL-1β and the type 1 interferon (IFN-β) response during the early time points of infection. Indeed, 4 hours post infection, IL-1β and IFN-β were both upregulated in *Slc27a1*-/- macrophages compared to scrambled controls consistent with their pro-inflammatory phenotype (*Figure 6—figure supplement 4A and B*). On the contrary, *Plin2*-/- macrophages downregulated IL-1β (*Figure 6—figure supplement 4B*). These data indicate that macrophages respond quite divergently to the deletion of

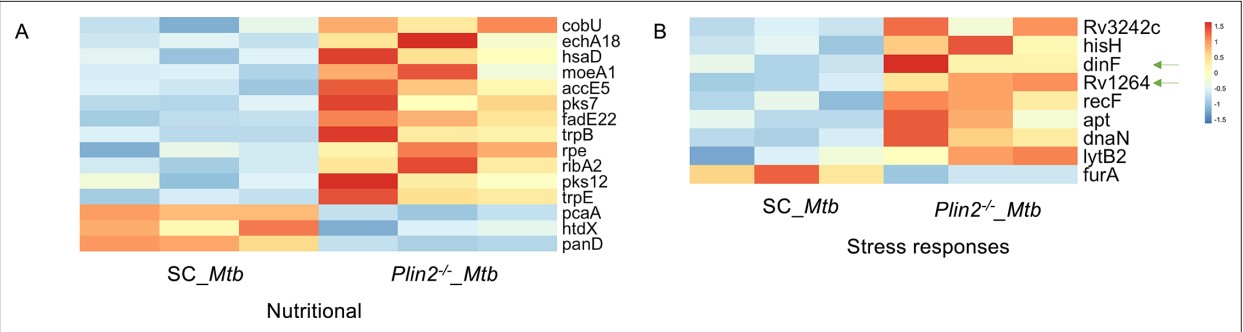

**Figure 7.** Nutritional and oxidative stress define the core transcriptome response of *Mycobacterium tuberculosis* (*Mtb*) inside *Plin2*⁻/⁻ macrophages. Heatmaps of nutritional (**A**) and oxidative stress (**B**) differentially expressed (DE) genes in *Plin2*⁻/⁻ macrophages. Arrows show genes that are also DE in *Cpt2*⁻/⁻ macrophages in a similar trend (***Supplementary file 4***).

The online version of this article includes the following source data and figure supplement(s) for figure 7:

**Figure supplement 1.** Total cellular reactive oxygen species (ROS) in *Plin2*⁻/⁻, *Slc27a1*⁻/⁻, and *Cpt2*⁻/⁻ macrophages.

**Figure supplement 1—source data 1.** Numerical source data for ***Figure 7—figure supplement 1***.

the different steps in fatty acid uptake, which implies that the intracellular pressures to which *Mtb* is exposed may also differ.

## Oxidative stress and nutrient limitation are major stresses experienced by *Mtb* in *Plin2*⁻/⁻ and *Cpt2*⁻/⁻ macrophages

We also analyzed transcriptomes from intracellular *Mtb* from scrambled controls, *Slc27a1*⁻/⁻, *Cpt2*⁻/⁻, and *Plin2*⁻/⁻ macrophages in parallel with host transcriptomes in ***Figure 5A***. Using an adjusted p-value of <0.1 and an absolute log$_2$ fold change >1.4, 0 genes were DE in *Slc27a1*⁻/⁻ macrophages, 105 *Mtb* genes were DE in *Plin2*⁻/⁻ macrophages (69 up, 36 down), and 10 genes were DE in *Cpt2*⁻/⁻ macrophages (3 up, 7 down) (***Supplementary file 4***). Despite being restrictive to *Mtb* growth (***Figure 1B***) and appearing more pro-inflammatory (***Figure 6A***), *Slc27a1*⁻/⁻ macrophages did not elicit a detectable shift in the transcriptional response in *Mtb* compared to control host cells. We speculate that pro-inflammatory responses in *Slc27a1*⁻/⁻ macrophages could be enough to restrict the growth of bacteria, but the resulting compensatory responses as evidenced by the upregulation of macrophage scavenger receptors (***Figure 6—figure supplement 1A***) alleviate some of the stresses that a lack of fatty acid import could be duly exerting on the bacteria. *Plin2*⁻/⁻ macrophages appeared to elicit the strongest transcriptional response from *Mtb,* which is consistent with our CFU data (***Figure 1B***) as *Plin2*⁻/⁻ macrophages exhibited the strongest growth restriction. Among the DE genes in *Mtb* from *Plin2*⁻/⁻ macrophages (***Supplementary file 4***), a significant number of upregulated genes are involved in nutrient assimilation (***Figure 7A***). *Mtb* in *Plin2*⁻/⁻ macrophages upregulated CobU (Rv0254c), which is predicted to be involved in the bacteria's cobalamin (vitamin B$_{12}$) biosynthesis. Vitamin B$_{12}$ is an important cofactor for the activity of *Mtb* genes required for cholesterol and fatty acid utilization (***Campos-Pardos et al., 2024***; ***Savvi et al., 2008***). Genes involved in de novo long-chain fatty acid synthesis (AccE5, Rv281) (***Bazet Lyonnet et al., 2014***), cholesterol breakdown (HsaD, Rv3569c) (***Lack et al., 2010***), β-oxidation of fatty acids (EchA18, Rv3373; FadE22, Rv3061c) (***Schnappinger et al., 2003***), purine salvage (Apt, Rv2584c) (***Warner et al., 2014***), and tryptophan metabolism (***Lott, 2020***) were also upregulated in *Mtb* from *Plin2*⁻/⁻ macrophages (***Figure 7A***). This metabolic realignment response is seen most frequently under nutrient-limiting conditions (***Huang et al., 2018***; ***Pisu et al., 2020***; ***Theriault et al., 2022***). *Mtb* in *Plin2*⁻/⁻ macrophages also appears to experience a significant level of other cellular stresses as genes involved in DNA synthesis and repair, general response to oxidative stress and pH survival in the phagosome (DnaN, Rv0002; RecF, Rv0003; DinF, Rv2836c; Rv3242c, Rv1264) were upregulated (***Figure 7B***). Among the downregulated genes in *Mtb* in *Plin2*⁻/⁻ macrophages, FurA (Rv1909c), a KatG repressor was the most significant (***Figure 7B***, ***Supplementary file 4***). FurA downregulation derepresses the catalase peroxidase, KatG, which promotes *Mtb* survival in oxidative stress conditions (***Zahrt et al., 2001***). These data suggest that *Plin2*⁻/⁻ macrophages could be, in part, restricting *Mtb* growth by increasing the production of ROS. The data also suggest that,

contrary to a previous report (**Knight et al., 2018**), blocking lipid droplet formation in host macrophages does place increased nutritional and oxidative stress on intracellular *Mtb*.

*Cpt2*[-/-] macrophages elicited a modest shift in transcriptional response from *Mtb*, and the majority of the DE genes (8 out of 10) were also DE in *Plin2*[-/-] macrophages (**Supplementary file 4**). This is in agreement with the host transcription response as *Cpt2*[-/-] and *Plin2*[-/-] macrophages share similar candidate antibacterial responses (**Figure 5A and B**, **Figure 6**, **Figure 6—figure supplement 3**). In common with the *Mtb* transcriptome response in *Plin2*[-/-] macrophages, upregulated genes in *Mtb* isolated from *Cpt2*[-/-] macrophages included those involved in response to oxidative stress (DinF, Rv2836c) (**Figure 7B**, **Supplementary file 4**). To substantiate some of these pathways, we assessed the levels of total cellular ROS in *Slc27a1*[-/-], *Plin2*[-/-], and *Cpt2*[-/-] macrophages in both *Mtb*-infected and uninfected conditions by staining the cells with the Invitrogen CellROX dye and confocal microscopy analysis of live stained cells. In both infected and uninfected conditions, all the three mutants displayed significantly elevated total cellular ROS compared to scramble controls (**Figure 7—figure supplement 1A and B**).

## Inhibitors of lipid metabolism block intracellular growth of *Mtb* in macrophages but not in broth culture

We next examined if pharmacological inhibitors would phenocopy the growth inhibition phenotypes we observed with specific gene deletions. Compounds that modulate lipid homeostasis are currently being investigated for HDT against TB, which is an area of considerable interest (**Kim et al., 2020**). Such compounds include metformin, a widely used antidiabetic drug that activates AMPK, inhibits

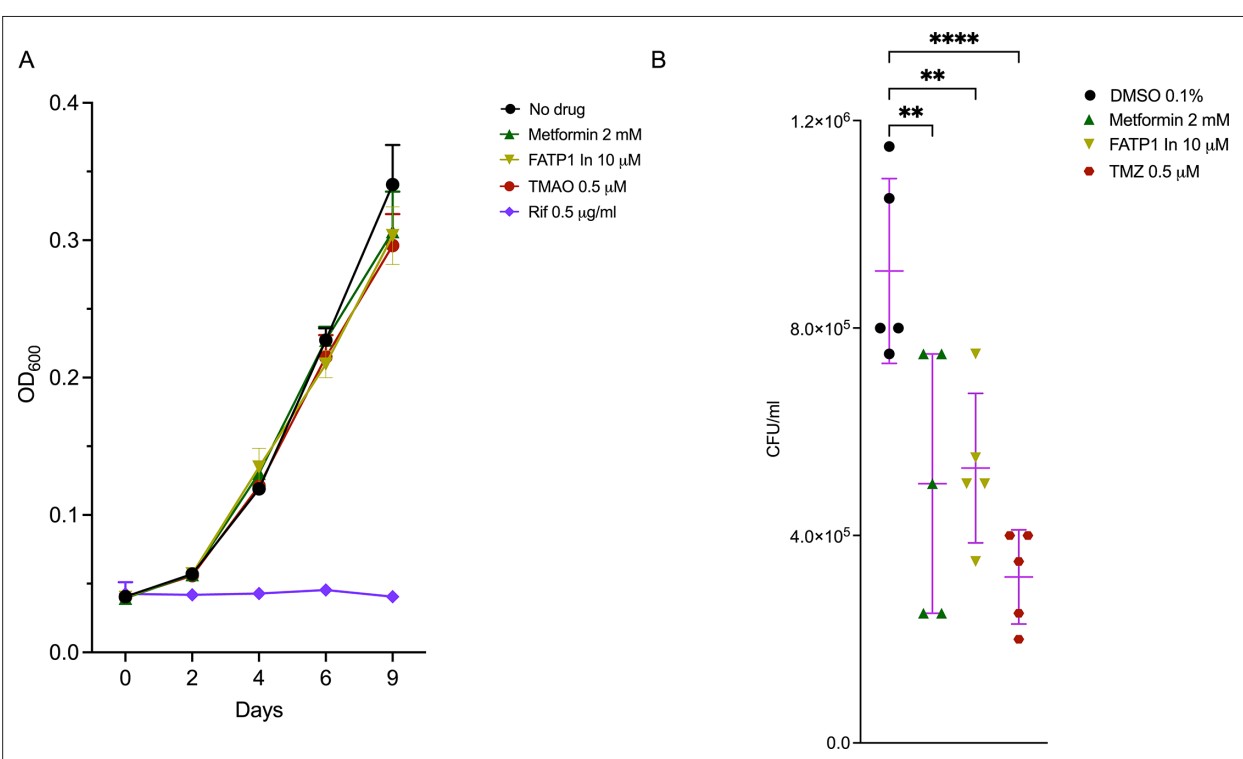

**Figure 8.** Inhibitors of fatty acid transport and metabolism block intracellular growth of *Mycobacterium tuberculosis* (*Mtb*) in macrophages. (**A**) Growth of *Mtb* in liquid broth in the absence of drug (DMSO) or presence of metformin, SLC27A1 inhibitor (FATP1 In, 10 µM) and the β-oxidation of fatty acid inhibitor, trimetazidine (TMZ, 500 nM). *Mtb* Erdman was grown to log phase and diluted to OD$_{600}$ 0.01 in 7H9 media in the presence of the above inhibitors. Growth kinetics were monitored by OD$_{600}$ measurements using a plate reader. Rifampicin (RIF) at 0.5 µg/ml was used as a total killing control. (**B**) Scramble macrophages were infected with *Mtb* Erdman at MOI 0.5. Inhibitors were added 3 hours post infection following which CFUs were plated 4 days post infection. n = 5 biological replicates; **p<0.01; ****p<0.0001, one-way ANOVA alongside Dunnett's multiple comparison test. Data are presented as mean values ± SD.

The online version of this article includes the following source data for figure 8:

**Source data 1.** Numerical source data for **Figure 8A and B**.

fatty acid synthesis, and promotes β-oxidation of fatty acids (*Fullerton et al., 2013*; *Singhal et al., 2014*). Chemical inhibition of fatty acid β-oxidation is already known to promote macrophage control of *Mtb* (*Chandra et al., 2020*; *Huang et al., 2018*). We targeted macrophage lipid homeostasis with trimetazidine (TMZ), an inhibitor of β-oxidation of fatty acids, metformin, and an SLC27A1 inhibitor, FATP1 In (*Matsufuji et al., 2013*). We assessed the impact of these compounds on extracellular *Mtb* cultured in broth over 9 days in the presence of the inhibitors (DMSO, TMZ; 500 nM, metformin; 2 mM, FATP1 In; 10 μM, rifampicin; 0.5 μg/ml). None of the three lipid metabolism inhibitors had a measurable effect on *Mtb* growth in liquid culture media compared to DMSO controls (*Figure 8A*). Treatment with rifampicin completely blocked bacterial growth under the same conditions (*Figure 8A*). We next infected scrambled sgRNA control macrophages with *Mtb* at MOI 0.5. Inhibitors were added to infected macrophages 3 hours post infection, and bacterial CFUs were enumerated 4 days post treatment. Consistent with previous observations (*Chandra et al., 2020*; *Singhal et al., 2014*), TMZ and metformin significantly reduced bacterial loads in macrophages compared to DMSO controls (*Figure 8B*). Similarly, FATP1 In also impacted the intracellular growth of *Mtb* (*Figure 8B*). The results provide independent data that both genetic and chemical modulation of fatty acid metabolism at different steps in the process negatively impact intracellular growth of *Mtb*.

## Discussion

It is clearly established that host-derived fatty acids and cholesterol are important carbon sources for *Mtb* (*Wilburn et al., 2018*). However, the relationship between *Mtb* and the infected host cell lipid metabolism remains a subject of conjecture. In this report, we characterized the role of macrophage lipid metabolism on the intracellular growth of *Mtb* by a targeted CRISPR-mediated knockout of host genes involved in fatty acid import, sequestration, and catabolism. Macrophage fatty acid uptake is mediated by the scavenger receptor CD36 and specialized long-chain fatty acid transporters, SLC27A1 and SLC27A4 (*Deng et al., 2023*). Earlier studies reported that a deficiency of CD36 enhances macrophage control of *Mtb,* albeit to a modest degree (*Hawkes et al., 2010*). Work from *Hawkes et al., 2010* indicated that the antimicrobial effectors in CD36-deficient macrophages were not due to bacterial uptake deficiencies, differences in the rate of *Mtb*-induced host cell death, and production of ROS or pro-inflammatory cytokines (*Hawkes et al., 2010*). We similarly observed a moderate *Mtb* growth restriction phenotype in our CRISPR-generated *Cd36*[-/-] macrophages. However, a strong growth restriction of *Mtb* was observed when we knocked out the long-chain fatty acid transporter, SLC27A1. *Slc27a1*[-/-] macrophages displayed altered metabolism characterized by the stabilization of HIF1α, activated AMPK, increased glycolysis, and reduced mitochondrial functions. Given that both CD36 and SLC27A1 perform similar functions, it is expected that there should be some degree of compensation between the transporters when either of the genes are deleted. Indeed, we found out that SLC27A1 knockout resulted in the upregulation of other lipid import transporters (MSR1, ABCC1). This would be consistent with the moderate anti-*Mtb* phenotypes in *Cd36*[-/-] macrophages that could easily be compensated by the presence of *long-chain fatty acid transporters* to alleviate the reduction in fatty acid supply experienced by intracellular *Mtb*. However, the *Slc27a1*[-/-] macrophage phenotype appears to be more severe on *Mtb* and could be exacerbated by an elevated pro-inflammatory response as has been reported previously both in vitro and in vivo (*Johnson et al., 2016*).

After uptake into the cells, most fatty acids either undergo β-oxidation in the mitochondria to provide energy or are esterified with glycerol phosphate to form triacylglycerols that may be incorporated into lipid droplets in the endoplasmic reticulum (*Deng et al., 2023*). Over the last two decades, it has been believed that lipid droplets are a nutrient source for *Mtb* in macrophages (*Russell et al., 2009*; *Daniel et al., 2011*; *Peyron et al., 2008*; *Singh et al., 2012*). However, recent work indicates lipid droplets may serve as centers for the production of pro-inflammatory markers and antimicrobial peptides (*Knight et al., 2018*; *Bosch et al., 2020*). It has also been reported that bone marrow macrophages derived from *Plin2*[-/-] mice did not have defects in the formation of lipid droplets and supported robust intracellular *Mtb* replication (*Knight et al., 2018*). However, our mutant macrophages with myeloid-specific knockout of PLIN2 are unable to form lipid droplets and are defective in supporting the growth of *Mtb*. The discrepancies with previous observations (*Knight et al., 2018*) could be a consequence of compensatory responses to PLIN2 knockout in whole mice, which when performed at embryonic level would allow for sufficient time for the cells to recover by upregulating related PLIN isoforms. In fact, *Plin2*[-/-] macrophages displayed the strongest anti-*Mtb* phenotypes amongst our

mutants exhibiting activated AMPK, increased glycolysis and autophagy, and impaired mitochondrial functions. *Mtb* isolated from *Plin2*[-/-] macrophages displayed signatures of severe nutrient limitation and oxidative stress damage.

It has also been previously reported that chemical, genetic, or miR33-mediated blockade of fatty acid β-oxidation in macrophages induces lipid droplet formation (*Chandra et al., 2020*; *Ouimet et al., 2016*) but that this enhanced lipid droplet formation does not correlate with improved intracellular *Mtb* growth (*Chandra et al., 2020*). We observed a similar phenotype as *Cpt2*[-/-] macrophages that generated larger and more abundant lipid droplets than scrambled control macrophages were still restrictive to *Mtb* growth. This implies that the presence or absence of lipid droplets does not in itself indicate whether a macrophage will support or restrict *Mtb* growth, and that the antimicrobial environment extends beyond simple nutrient availability.

In summary, our study shows that blocking macrophage's ability to import, sequester, or catabolize fatty acids chemically or by genetic knockout impairs *Mtb* intracellular growth. There are shared features between potential antimicrobial effectors in macrophages that lack the ability to import (*Slc27a1*[-/-]) or metabolize fatty acids (*Plin2*[-/-], *Cpt2*[-/-]) such as increased glycolysis, stabilized HIF1α, activated AMPK, enhanced autophagy, and production of ROS. However, there are also intriguing points of divergence as *Slc27a1*[-/-] macrophages are more pro-inflammatory while *Plin2*[-/-] macrophages appear to be broadly anti-inflammatory. The routes to *Mtb* growth restriction in these mutant macrophages are clearly more complex than the bacteria's inability to acquire nutrients. The data further emphasizes that targeting fatty acid homeostasis in macrophages at different steps in the process (uptake, storage, and catabolism) is worthy of exploring in the development of new therapeutics against TB.

## Materials and methods

All materials and methods are as described (*Simwela et al., 2024*) unless otherwise specified.

### Flow cytometry and western blot analysis

Generation of CRISPR mutant Hoxb8 macrophages was carried out as described previously (*Simwela et al., 2024*). Antibodies used for both western blot and flow cytometry were as follows: rat anti-mouse CD36:Alexa Fluor647 (Bio-Rad, 10 µl/million cells), rabbit anti-PLIN2 (Proteintech, 1:1000), rabbit anti-SLC27A1 (Affinity Biosciences, 1:1000), rabbit anti-CPT1A antibody (Proteintech, 1:1000), rabbit anti-CPT2 antibody (Proteintech, 1:1000), rabbit anti-HIF1α antibody (Proteintech, 1:1000), rabbit anti-AMPKα (1:1000, Cell Signalling Technology), rabbit anti-Phospho-AMPKα (1:500, Cell Signalling Technology), rabbit anti-LC3B (1:1000, Cell Signalling Technology), and rabbit anti-β-actin (1:1000, Cell Signalling Technology). For western blots, secondary antibodies used were anti-rabbit/mouse StarBright Blue 700 (1:2500, Bio-Rad). Blots were developed and imaged as described previously (*Simwela et al., 2024*).

### Staining for cellular lipid droplets

Macrophages monolayers in Ibidi eight-well chambers were supplemented with exogenous 400 µM oleate for 24 hours to induce the formation of lipid droplets (*Listenberger and Brown, 2007*). Cells were then fixed in 4% paraformaldehyde and stained with BODIPY 493/503 (Invitrogen, 1 µg/ml) in 150 mM sodium chloride. Stained cells were mounted with media containing DAPI and imaged using a Leica SP5 confocal microscope.

### Seahorse XF palmitate oxidation stress test

A modified Seahorse mitochondrial stress test was used to measure macrophage's ability to oxidize palmitate in substrate-limiting conditions. Two days before the assay, $1 \times 10^5$ cells were plated in Seahorse cell culture mini plates. One day before the assay, macrophage media was replaced with the Seahorse substrate-limited growth media (DMEM without pyruvate supplemented with 0.5 mM glucose, 1 mM glutamine, 1% FBS, and 0.5 mM L-carnitine). On the day of the assay, substrate-limited media was replaced with assay media (DMEM without pyruvate supplemented with 2 mM glucose and 0.5 mM L-carnitine). In selected treatment conditions, cells were either supplied with BSA, BSA

palmitate, or BSA palmitate plus etomoxir (4 μM). OCRs were measured using the Mito Stress Test assay conditions as described previously (*Simwela et al., 2024*).

## Rescue of the *Mtb* Δicl1 mutant in oleate supplemented media

The *Mtb* H37Rv Δicl1 mutant expressing mCherry (*Lee et al., 2013*) was used for the rescue experiments. The strain was maintained in 7H9 OADC broth as previously described (*Simwela et al., 2024*) in the presence of kanamycin (25 μg/ml) and hygromycin (50 μg/ml). 24 hours before infection, macrophages were cultured in normal macrophage media or media supplemented with 400 μM oleate. Cells were then infected with the *Mtb* Δicl1 mutant at MOI 5. The bacterial mCherry signal was measured on day 0 and day 5 post infection on an Envision plate reader (PerkinElmer). Oleate was maintained throughout the experiment in the rescue assay conditions.

## Measurement of total cellular ROS

Uninfected or *Mtb*-infected macrophages monolayers in Ibidi eight-well chambers were stained with the CellROX Deep Red dye (Invitrogen) as per the manufacturer's staining protocol. Live stained cells were imaged on a Leica SP5 confocal microscope. Z-stacks were reconstructed in ImageJ from which mean fluorescence intensities (MFIs) for individual cells were obtained.

## Acknowledgements

We would like to thank Dr. Jen K Grenier and Ann E Tate from the Cornell BRC Transcriptional Regulation and Expression Facility for their help with the development of dual RNA-Seq protocols. This work was supported by grants from the National Institutes of Health (AI155319, AI162598, and OD032135), Bill and Melinda Gates Foundation, and the Mueller Health Foundation to DGR. EJ was supported by T32AI007349.

## Additional information

### Funding

| Funder | Grant reference number | Author |
| --- | --- | --- |
| National Institute of Allergy and Infectious Diseases | AI155319 | David G Russell |
| National Institute of Allergy and Infectious Diseases | AI162598 | David G Russell |
| NIH Office of the Director | OD032135 | David G Russell |
| National Institute of Allergy and Infectious Diseases | T32AI007349 | Eleni Jaecklein |
| Mueller Health Foundation | | David G Russell |

The funders had no role in study design, data collection and interpretation, or the decision to submit the work for publication.

### Author contributions

Nelson V Simwela, Conceptualization, Data curation, Formal analysis, Investigation, Methodology, Validation, Visualization, Writing – original draft, Writing – review and editing; Eleni Jaecklein, Visualization; Christopher M Sassetti, Resources, Visualization; David G Russell, Conceptualization, Supervision, Funding acquisition, Project administration, Writing – review and editing

### Author ORCIDs

Nelson V Simwela ● https://orcid.org/0000-0002-4734-0518
Christopher M Sassetti ● https://orcid.org/0000-0001-6178-4329
David G Russell ● https://orcid.org/0000-0002-9748-750X

### Ethics

All of our protocols were reviewed and approved by Institutional Animal Care and User Committee of Cornell University, protocol # 2011-0086.

Reviewer #1 (Public review): https://doi.org/10.7554/eLife.102980.3.sa1
Reviewer #2 (Public review): https://doi.org/10.7554/eLife.102980.3.sa2
Reviewer #3 (Public review): https://doi.org/10.7554/eLife.102980.3.sa3
Author response https://doi.org/10.7554/eLife.102980.3.sa4

## Additional files

### Supplementary files

Supplementary file 1. sgRNAs primers and ICE scores for lipid import and metabolism gene targets in this study.

Supplementary file 2. Differentially expressed genes in *Plin2*[-/-], *Slc27a1*[-/-], and *Cpt2*[-/-] *Mtb*-infected macrophages.

Supplementary file 3. Enriched GO terms in upregulated and downregulated genes in *Plin2*[-/-], *Slc27a1*[-/-], and *Cpt2*[-/-] *Mtb*-infected macrophages.

Supplementary file 4. *Mtb* differentially expressed genes in *Plin2*[-/-] and *Cpt2*[-/-] infected macrophages.

MDAR checklist

### Data availability

The RNA-seq data from the dual RNA-seq analysis of infected mouse macrophages, which includes both macrophage and Mtb reads, are available in GEO (GSE270571).

The following dataset was generated:

| Author(s) | Year | Dataset title | Dataset URL | Database and Identifier |
|---|---|---|---|---|
| Simwela NV, Jaecklein E, Sassetti CM, Russell DG | 2024 | Impaired fatty acid import and metabolism in macrophages restricts intracellular growth of *Mycobacterium tuberculosis* | https://www.ncbi.nlm.nih.gov/geo/query/acc.cgi?acc=GSE270571 | NCBI Gene Expression Omnibus, GSE270571 |

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
