## [Editor Report · eLife Assessment]

This **important** study reveals that disrupting fatty acid metabolism in macrophages significantly restricts the growth of *Mycobacterium tuberculosis*, showing that impaired lipid processing triggers various antimicrobial responses. Overall, the approach is robust utilizing CRISPR-Cas9 knockout of multiple genes involved in lipid metabolism that yielded **convincing** data. This work highlights how host lipid metabolism affects the ability of tubercle bacilli to thrive intracellularly, pointing to potential new therapeutic targets.

---

## [Referee Report · Reviewer #1 (Public review)]

Summary:

This study investigates the role of macrophage lipid metabolism in the intracellular growth of *Mycobacterium tuberculosis*. By using a CRISPR-Cas9 gene-editing approach, the authors knocked out key genes involved in fatty acid import, lipid droplet formation, and fatty acid oxidation in macrophages. Their results show that disrupting various stages of fatty acid metabolism significantly impairs the ability of Mtb to replicate inside macrophages. The mechanisms of growth restriction included increased glycolysis, oxidative stress, pro-inflammatory cytokine production, enhanced autophagy, and nutrient limitation. The study demonstrates that targeting fatty acid homeostasis at different stages of the lipid metabolic process could offer new strategies for host-directed therapies against tuberculosis.

The work is convincing and methodologically strong, combining genetic, metabolic, and transcriptomic analyses to provide deep insights into how host lipid metabolism affects bacterial survival.

Strengths:

The study uses a multifaceted approach, including CRISPR-Cas9 gene knockouts, metabolic assays, and dual RNA sequencing, to assess how various stages of macrophage lipid metabolism affect Mtb growth. The use of CRISPR-Cas9 to selectively knock out key genes involved in fatty acid metabolism enables precise investigation of how each step-lipid import, lipid droplet formation, and fatty acid oxidation-affects Mtb survival. The study offers mechanistic insights into how different impairments in lipid metabolism lead to diverse antimicrobial responses, including glycolysis, oxidative stress, and autophagy. This deepens the understanding of macrophage function in immune defense.

The use of functional assays to validate findings (e.g., metabolic flux analyses, lipid droplet formation assays, and rescue experiments with fatty acid supplementation) strengthens the reliability and applicability of the results.

By highlighting potential targets for HDT that exploit macrophage lipid metabolism to restrict Mtb growth, the work has significant implications for developing new tuberculosis treatments.

Weaknesses:

The experiments were primarily conducted in vitro using CRISPR-modified macrophages. While these provide valuable insights, they may not fully replicate the complexity of the in vivo environment where multiple cell types and factors influence Mtb infection and immune responses. Yet, I agree that the Hoxb8 in vitro model provides a powerful genetic tool to interrogate host-Mtb interactions using primary macrophages that represent the bone marrow-derived macrophage lineage, instead of using cell lines.

Comments on revisions: The authors have addressed my comment satisfactorily.

---

## [Referee Report · Reviewer #2 (Public review)]

Summary:

Host-derived lipids are an important factor during Mtb infection. In this study, using CRISPR knockouts of genes involved in fatty acid uptake and metabolism, the authors claim that a compromised uptake, storage or metabolism of fatty acid in the hosts restricts Mtb growth upon infection. The mechanism involves increased glycolysis, autophagy, oxidative stress, pro-inflammatory cytokines and nutrient limitation. The study may be useful for developing novel host-directed approaches against TB.

Strengths:

The study's strength is the use of clean HOXB8-derived primary mouse macrophage lines for generating CRISPR knockouts.

Weaknesses:

The strength of evidence on autophagy and redox stress remains incomplete.

Comments on revisions:

The authors have revised the manuscript and addressed some of the earlier concerns. However, some of the interpretations and responses are incorrect.

Overall, the level of evidence to state the following in the abstract- ‘Our analyzes demonstrate that macrophages which cannot either import, store or catabolize fatty acids restrict Mtb growth by both common and divergent anti-microbial mechanisms, including increased glycolysis, increased oxidative stress, production of pro-inflammatory cytokines, enhanced autophagy and nutrient limitation’ is incomplete.

There is an increase in glycolysis and pro-inflammatory cytokines and, to some extent, oxidative stress. The same can not be said about autophagy. Unfortunately, the authors did not try to establish a direct role of any of these pathways in restricting bacterial growth in the absence of any of the three genes studied.

Major concern:

Autophagy: The LC3 WB does not, by any stretch of the imagination, convince that there is an increase in autophagy flux, as inferred by the authors. Authors correctly cite the ‘Guidelines to autophagy’ paper. Unfortunately, they cite it only selectively to justify their assessment. The LC3II/LC3I ratio indicates the number of autophagosomes present. This ratio can also increase if there is an active block of autophagosome maturation. That's why having BafA1 or CQ controls is important to assess the active autophagosome maturation. However, the authors sidestep this serious consideration by claiming some ‘pleiotropic impact on Mtb’. With BafA1 and CQ, the only assay one needs is to measure the impact on LC3II levels. In the absence of this assay, the evidence supporting the role of autophagy is incomplete.

The main concern regarding autophagy results is that autophagy induction can typically bring down oxidative stress and classically has anti-inflammatory outlay. Thus, increased glycolysis, inflammatory cytokine production and redox stress indicate more towards a potential block in autophagy at the maturation step. This necessitates validation using autophagy flux assays.

Oxidative stress: Showing a representative image for the corresponding representative groups would be more convincing. For example, there is no clarity on whether, in the infected group, there was any staining for Mtb to analyse only the infected cells.

---

## [Referee Report · Reviewer #3 (Public review)]

Summary:

This study provides significant insights into how host metabolism, specifically of lipids, influences the pathogenesis of *Mycobacterium tuberculosis* (Mtb). It builds on existing knowledge about Mtb's reliance on host lipids and emphasizes the potential of targeting fatty acid metabolism for therapeutic intervention.

Strengths:

To generate the data, the authors use CRISPR technology to precisely disrupt the genes involved in lipid import (CD36, FATP1), lipid droplet formation (PLIN2) and fatty acid oxidation (CPT1A, CPT2) in mouse primary macrophages. The Mtb Erdman strain is used to infect the macrophage mutants. The study, revealsspecific roles of different lipid-related genes. Importantly, results challenge previous assumptions about lipid droplet formation and show that macrophage responses to lipid metabolism impairments are complex and multifaceted. The experiments are well-controlled and the data is convincing.

Overall, this well-written paper makes a meaningful contribution to the field of tuberculosis research, particularly in the context of host-directed therapies (HDTs). It suggests that manipulating macrophage metabolism could be an effective strategy to limit Mtb growth.

Weaknesses:

None noted. The manuscript provides important new knowledge that will lead mpvel to host-directed therapies to control Mtb infections.

Comments on revisions: The authors have addressed the concerns of the reviewers.

---

## [Author Response]

The following is the authors’ response to the original reviews.

**Public Reviews:**

**Reviewer #1 (Public review):**
Summary:This study investigates the role of macrophage lipid metabolism in the intracellular growth of *Mycobacterium tuberculosis*. By using a CRISPR-Cas9 gene-editing approach, the authors knocked out key genes involved in fatty acid import, lipid droplet formation, and fatty acid oxidation in macrophages. Their results show that disrupting various stages of fatty acid metabolism significantly impairs the ability of Mtb to replicate inside macrophages. The mechanisms of growth restriction included increased glycolysis, oxidative stress, pro-inflammatory cytokine production, enhanced autophagy, and nutrient limitation. The study demonstrates that targeting fatty acid homeostasis at different stages of the lipid metabolic process could offer new strategies for host-directed therapies against tuberculosis.The work is convincing and methodologically strong, combining genetic, metabolic, and transcriptomic analyses to provide deep insights into how host lipid metabolism affects bacterial survival.Strengths:The study uses a multifaceted approach, including CRISPR-Cas9 gene knockouts, metabolic assays, and dual RNA sequencing, to assess how various stages of macrophage lipid metabolism affect Mtb growth. The use of CRISPR-Cas9 to selectively knock out key genes involved in fatty acid metabolism enables precise investigation of how each step-lipid import, lipid droplet formation, and fatty acid oxidation affect Mtb survival. The study offers mechanistic insights into how different impairments in lipid metabolism lead to diverse antimicrobial responses, including glycolysis, oxidative stress, and autophagy. This deepens the understanding of macrophage function in immune defense.The use of functional assays to validate findings (e.g., metabolic flux analyses, lipid droplet formation assays, and rescue experiments with fatty acid supplementation) strengthens the reliability and applicability of the results.By highlighting potential targets for HDT that exploit macrophage lipid metabolism to restrict Mtb growth, the work has significant implications for developing new tuberculosis treatments.Weaknesses:The experiments were primarily conducted in vitro using CRISPR-modified macrophages. While these provide valuable insights, they may not fully replicate the complexity of the in vivo environment where multiple cell types and factors influence Mtb infection and immune responses.

We thank the reviewer for pointing this out. We acknowledge that our in vitro system may indeed not fully replicate the complex in vivo environment given of what is becoming to light of macrophage heterogenous responses to Mtb infection in whole animal models. We do believe, however, that the Hoxb8 in vitro model provides a powerful genetic tool to interrogate host-Mtb interactions using primary macrophages that represent the bone marrow-derived macrophage lineage.

**Reviewer #2 (Public review):**
Summary:Host-derived lipids are an important factor during Mtb infection. In this study, using CRISPR knockouts of genes involved in fatty acid uptake and metabolism, the authors claim that a compromised uptake, storage, or metabolism of fatty acid restricts Mtb growth upon infection. Further, the authors claim that the mechanism involves increased glycolysis, autophagy, oxidative stress, pro-inflammatory cytokines, and nutrient limitation. The authors also claim that impaired lipid droplet formation restricts Mtb growth. However, promoting lipid droplet biogenesis does not reverse/promote Mtb growth.Strengths:The strength of the study is the use of clean HOXB8-derived primary mouse macrophage lines for generating CRISPR knockouts.Weaknesses:There are many weaknesses of this study, they are clubbed into four categories below(1) Evidence and interpretations: The results shown in this study at several places do not support the interpretations made or are internally contradictory or inconsistent. There are several important observations, but none were taken forward for in-depth analysis.a) The phenotypes of PLIN2^-/-^, FATP1^-/-^, and CPT-/- are comparable in terms of bacterial growth restriction; however, their phenotype in terms of lipid body formation, IL1B expression, etc., are not consistent. These are interesting observations and suggest additional mechanisms specific to specific target genes; however, clubbing them all as altered fatty acid uptake or catabolism-dependent phenotypes takes away this important point.

We thank the reviewer for highlighting this. Our focus was on assessing the impact of manipulating lipid homeostasis in macrophages at several stages and the consequences this has on the intracellular growth of Mtb. Throughout the manuscript (abstract, results and discussion), we have continuously emphasized that interfering with lipid handling at several stages in macrophages results in both conserved and divergent antimicrobial responses against intracellular Mtb.

b) Finding the FATP1 transcript in the HOXB8-derived FATP1^-/-^ CRISPR KO line is a bit confusing. There is less than a two-fold decrease in relative transcript abundance in the KO line compared to the WT line, leaving concerns regarding the robustness of other experiments as well using FATP1^-/-^ cells.

CRISPR-Cas9 targeting of genes with single sgRNAs as is the case with our mutants generates insertions and deletions (INDELs) at the CRISPR cut site. These INDELs do not block mRNA transcription totally, and this is widely reported in the field. Because of this, quantitative RT-PCR or RNA-seq methods are not routinely used to verify CRISPR knockouts as they are not sensitive enough to identify INDELs. We provide INDEL quantification and knockout efficiencies by ICE analysis in supplemental file 1 for all the mutants used in the study. We also demonstrate protein depletion by western blot and flow cytometry for all the mutants (Figure 1 - figure supplement 1). Only mutants with greater than >90% protein depletion were used for subsequent characterization.

c) No gene showing differential regulation in FATP^-/-^ macrophages, which is very surprising.

We assume the reviewer is referring to the Mtb transcriptome response in FATP1^-/-^ macrophages, which we agree was unexpected. However, we saw a significant compensatory response in the host cell (at transcriptional level) in FATP1^-/-^ macrophages as evidenced by an upregulation of other fatty acid transporters (Figure 5 - figure supplement 1, now Figure 6 - figure supplement 1). We believe that these compensatory responses could, in part, alleviate the stresses the bacteria experience within the cell. We discuss this point in the manuscript.

d) ROS measurements should be done using flow cytometry and not by microscopy to nail the actual pattern.

We thank the reviewer for the suggestion. However, confocal imaging is also widely used to measure ROS with similar quantitative power and individual cell resolution (PMID: 32636249, 35737799).

(2) Experimental design: For a few assays, the experimental design is inappropriatea) For autophagy flux assay, immunoblot of LC3II alone is not sufficient to make any interpretation regarding the state of autophagy. This assay must be done with BafA1 or CQ controls to assess the true state of autophagy.

We would like to point out that monitoring LC3I to LC3II conversion by western blot, confocal imaging of LC3 puncta and qPCR analysis of autophagy related genes are all validated assays for monitoring autophagic flux in a wide variety of cells. We refer the reviewer to the latest extensive guidelines on the subject (PMID: 33634751). Furthermore, Bafilomycin A and chloroquine are not specific inhibitors of autophagy and therefore are of limited value as controls. BafA is an inhibitor of the proton-ATPase apparatus and can indirectly impact autophagy through activity on the Ca-P60A/SERCA pathway. Chloroquine impacts vacuole acidification, autophagosome/lysosome fusion and slows phagosome maturation. So, while BafA and chloroquine will reduce autophagy; their effects are pleotropic and their impact on Mtb is unknown.

b) Similarly, qPCR analyses of autophagy-related gene expression do not reflect anything on the state of autophagy flux.

See our response above.

(3) Using correlative observations as evidence:a) Observations based on RNAseq analyses are presented as functional readouts, which is incorrect.

We are not entirely sure where we used our RNA-seq data sets as functional readouts. We used our transcriptome data to provide a preliminary identification of anti-microbial responses in the mutant macrophages infected with Mtb and we mention this at the beginning of the RNA-seq results sections. Where applicable, we followed up and confirmed the more compelling RNA-seq data either by metabolic flux analyzes, qPCR, ROS measurements, and quantitative imaging.

b) Claiming that the inability to generate lipid droplets in PLIN2^-/-^ cells led to the upregulation of several pathways in the cells is purely correlative, and the causal relationship does not exist in the data presented.

It was not our intention to infer causality. We have re-written the beginning of the sentence, and it now starts with ‘Meanwhile, Mtb infection of PLIN2^-/-^ macrophages led to upregulation’ which hopefully eliminates any association to causality.

(4) Novelty: A few main observations described in this study were previously reported. That includes Mtb growth restriction in PLIN2 and FATP1 deficient cells. Similarly, the impact of Metformin and TMZ on intracellular Mtb growth is well-reported. While that validates these observations in this study, it takes away any novelty from the study.

To the best of our knowledge, Mtb growth restrictions in PLIN2 and FATP1 deficient macrophages have not been reported elsewhere. To the contrary, PLIN2 knockout macrophages obtained from PLIN2 deficient mice have been reported to robustly support Mtb replication (PMID: 29370315). We extensively discuss these discrepancies in the manuscript. We also discuss and cite appropriate references where Mtb growth restriction for similar macrophage mutants have been reported (CD36^-/-^ and CPT2^-/-^). Our aim was to carry out a systematic myeloid specific genetic interference of fatty acid import, storage and catabolism to assess the effect on Mtb growth at all stages of lipid handling instead of focusing on one target. In the chemical approach, we used TMZ and Metformin deliberately because they had already been reported as being active against intracellular Mtb and we wished to place our data in the context of existing literature. These studies have been referenced extensively in the text.

(5) Manuscript organisation: It will be very helpful to rearrange figures and supplementary figures.

New figures have been added, and existing ones have been re-arranged where necessary. See our responses to recommendations for authors.

**Reviewer #3 (Public review):**
Summary:This study provides significant insights into how host metabolism, specifically lipids, influences the pathogenesis of *Mycobacterium tuberculosis* (Mtb). It builds on existing knowledge about Mtb's reliance on host lipids and emphasizes the potential of targeting fatty acid metabolism for therapeutic intervention.Strengths:To generate the data, the authors use CRISPR technology to precisely disrupt the genes involved in lipid import (CD36, FATP1), lipid droplet formation (PLIN2), and fatty acid oxidation (CPT1A, CPT2) in mouse primary macrophages. The Mtb Erdman strain is used to infect the macrophage mutants. The study, reveals specific roles of different lipid-related genes. Importantly, results challenge previous assumptions about lipid droplet formation and show that macrophage responses to lipid metabolism impairments are complex and multifaceted. The experiments are well-controlled and the data is convincing.Overall, this well-written paper makes a meaningful contribution to the field of tuberculosis research, particularly in the context of host-directed therapies (HDTs). It suggests that manipulating macrophage metabolism could be an effective strategy to limit Mtb growth.Weaknesses:None noted. The manuscript provides important new knowledge that will lead mpvel to host-directed therapies to control Mtb infections.
**Recommendations for the authors:**

**Reviewer #1 (Recommendations for the authors):**
The study presents compelling and well-supported conclusions based on a solid body of evidence. However, the clarity of several figures could be improved for better understanding.(1) In Figure 1, panels B and C are referenced incorrectly in the text.

We thank the reviewer for identifying the error. This has now been corrected

(2) Figures 2 and S2 would benefit from being combined or reorganized to display the data related to infected and uninfected cells together, making it easier for the reader to interpret.

We thank the reviewer for the suggestion. However, we believe that combining the two figures would further complicate the merged figure making it even more difficult to interpret. We decided to highlight the mutant macrophage’s responses upon Mtb infection in Figure 2 and put the uninfected data sets in supplementary information given that the OCR and ECAR trends were similar and as expected in both infected and uninfected states.

(3) Figure 3 is mislabeled, with four panels shown in the figure, but only panels A and B are mentioned in both the text and the figure legend.

We thank the reviewer for the observation. Figure 3 has been extensively revised. We have included new blots, statistical comparisons and a corresponding new supplementary figure (Figure 3 - figure supplement 1). We have verified that the figure panels are labelled correctly and appropriately referenced in the manuscript text.

(4) Figure 5 is overly complex and difficult to interpret. Simplifying the figure, possibly by reducing the amount of data or breaking it into more digestible parts, would enhance its readability.

We thank the reviewer for the suggestion. We have separated the figure into two parts which are now Figure 5 for the PCA and Venn diagrams and Figure 6 for the pathway enrichment figure panels. We have increased the resolution of both figures in the revised manuscript to improve readability.

(5) Panel 6A is not particularly informative and could either be omitted with a more detailed explanation provided in the text, or replaced with a clearer visual representation, such as Venn diagrams, to improve data visualization.

We thank the reviewer for the suggestion. We have removed Figure 6A given that detailed explanation of the panel is already available in the manuscript text.

(6) Additionally, on line 309, the word ‘to’ is missing before ‘generate’.

We thank the reviewer for identifying this. This sentence has now been re-written to address some unintended inferences of causation in line with recommendations from reviewer 2.

**Reviewer #2 (Recommendations for the authors):**
(1) Manuscript Organisations: The manuscript is very poorly organised. Supplemental figures are labelled very unconventionally, and that creates much confusion in following the manuscript. Some of the results in the supplementary figures could be easily kept in the main figures, as it is difficult to compare plots between the main figures and the supple figures. The results of RNAseq experiments are impossible to follow with very small fonts. Overall, the figures are very casually organised and can certainly be improved.

We would like to clarify that supplemental figures are labelled and organized as is in line with the eLife formatting of supplemental figures. We deliberately put some redundant figures like Figure 2 - figure supplement 1 in supplementary information (see our response to reviewer 1 recommendations on the same). We have split the RNA-seq Figure 5 into two separate figures (now Figure 5 and 6) and increased their resolution to improve readability.

(2) Figure 3: Among the KO lines, only PLIN2^-/-^ had a higher HIF1a level before infection. Infection surely leads to higher levels across the three cases.

We have generated replicate western blots and provide statistical quantitation for both HIF1a, AMPK and pAMPK. Figure 3 has now been revised extensively, replicate blots are in Figure 3 - figure supplement 1. We have updated the text to reflect the reviewer observation which was also consistent with our statistical quantification.

(3) pAMPK blots are of very poor quality. Without quantification, the trend mentioned in the text is not clearly visible.

We have provided two more replicate blots for AMPK/pAMPK and provide statistical quantification as described above.

(4) Line 230: Regarding autophagy flux, neither the data suggest what is interpreted nor is this experiment correctly done. LC3 WB and autophagy gene qPCR: Unfortunately, LC3 WB, the way it was done, does not tell anything about the state of autophagy in these cells. A very mild LC3II increase is noted in CPT2^-/-^ cells upon infection; the rest of the others do not show any change. This assay is not done correctly. To interpret LC3II WB, one needs to include the Bafilomycin A1 control, usually +Baf and -Baf run in the adjacent wells in the gel. Similarly, qPCR results are not indicative of any increase in autophagy. Regulation of ATG7, MAP1LC3B, and ULK1 is more at the post-translational level than the transcriptional level.

We have provided an additional replicate blot together with statistical quantification of LC3II/LC3I ratios in the revised Figure 3 - figure supplement 2. Our quantifications remain consistent with our prior assertations in the manuscript text. See our response in the public review section concerning autophagy assays and the use of Baf or chloroquine as controls.

(5) Exogenous oleate fails to rescue the Mtb icl1-deficient mutant in FATP1^-/-^, PLIN2^-/-^ and CPT2^-/-^ macrophages: this result is confusing. Lipid uptake and metabolism have been the central players so far; however, here, the phenotypes of FATP1 and CPT2 in terms of lipid body accumulation are very distinct. Therefore, the assessment that Mtb growth inhibition is due to factors other than limited access to fatty acid is not consistent with the theme of the study.

Nutrient limitation is a distinct transcriptional signature of Mtb, at least in PLIN2^-/-^ macrophages (Figure 7). We used the oleate supplementation assay with the Mtb Dicl1 mutant to assess whether nutrient restriction was the sole anti-microbial pathway against Mtb in the knockout macrophages. This would have been the case (to a certain extent) if the growth of the Mtb Dicl1 mutant was rescuable upon addition of exogenous oleate in the knockout macrophages. Our data clearly shows that this is not the case and that in addition to nutrient limitation, interference with lipid processing results in several other macrophage anti-microbial responses against the bacteria. We extensively discuss these points in the abstract, results and discussion sections of the manuscript.

(6) Line 309: ‘Meanwhile, inability generate lipid droplets in Mtb infected PLIN2^-/-^ macrophages led to upregulation in pathways involved in ribosomal biology, MHC class 1 antigen presentation, canonical glycolysis, ATP metabolic processes and type 1 interferon responses (Figure 5C, Supplementary file 3).’ This is just a correlative observation. However, it is mentioned here as a causal mechanism.

We have revised this sentence to remove any unintended inference of causation.

(7) IL-1b is upregulated in FATP-/- macrophages, no effect in CPT2^-/-^ macrophages, but downregulated in PLIN2^-/-^ macrophages. Moreover, this effect is very transient, and by 24 hours, all these differences are lost. This suggests the mechanism of action, as their pro-bacterial function shown in Figure 1, is very distinct for different proteins, and FA metabolism is probably not the common denominator across these phenotypes.

We agree with the reviewer, and we extensively discuss this in the manuscript text (results and discussion). Clearly, they are shared anti-microbial responses across the mutants, but they are also points of divergence. We would like to further clarify that pro-inflammatory responses (IL-1b or IFN-B) in Mtb infected macrophages show a biphasic early upregulation (up to 8 hours of infection) followed by a rapid resolution phase (24-48 hours post infection). This is well reported in the literature (PMID: 30914513). It is common for pro-inflammatory gene expression differences to be temporary lost during the resolution phase (PMID: 30914513, 39472457). IL-1b expression profiles return to the 4-hour equivalent profile in Mtb infected FATP1^-/-^ and PLIN2^-/-^ macrophages 4 days post infection (Figure 6A, Figure 6 - figure supplement 2B, Supplementary file 2)

(8) It is very surprising that FATP-/- macrophages do not show any change in Mtb gene expression. The robustness of this experiment and analysis appears doubtful, given that the phenotype in terms of bacterial growth was clean.

See our response to this comment in the public reviews section

(9) Figure 5, Supplementary Figure 1: Among the FA transporters, authors also show data for FATP1. I am surprised to see FATP1 expression levels in the FATP1^-/-^ cells. This puts into doubt every dataset using FATP-/- cells in this study.

See our response to this comment in the public reviews section

(10) Unfortunately, with the kind of evidence presented, it is far-fetched to claim that PLIN2^-/-^ macrophages restrict Mtb growth by increasing ROS production. There is no evidence for this statement. The MFI units in Figure 6, Supplementary 1 are too small to extract meaningful interpretations. Moreover, the data appears to be arrived at by combining multiple technical replicates. Usually, flow cytometry data are more reliable for CellROX assays. Microscopy is not the technique of choice for this assay.

We would like to point out that MFIs are arbitrary units set to predetermined reference points. In our case, the reference was background fluorescence in CellROX unstained cells and cells stained with CellROX equivalent fluorophore conjugated isotype antibodies. We are not entirely sure what the reviewer means by ‘small’ in these contexts. And the data is not entirely from technical replicates. Reported MFIs are from three independent repeats with MFI reads of at least 30 cells per replicate. We have added this clarification in Figure 6 - figure supplement 1 legend, now Figure 7 - figure supplement 1. See our response in the public reviews section on the use of confocal microcopy to image and quantify ROS. Furthermore, the Mtb transcriptional response in PLIN2^-/-^ and CPT2^-/-^ macrophages is clearly indicative of increased oxidative stresses (Figure 7).

(11) The CFU results with Metformin and TMZ are on the expected lines, as published earlier by others. FATP1 In data is good and aligned with the knockout phenotype.

We thank the reviewer for the note.

(12) Western blots, when interpreted for quantitative differences, must be quantified, and data should be represented as plots with statistical analysis.

Replicate blots have been provided and statistical quantifications performed.